# A neonatal mouse model characterizes transmissibility of SARS-CoV-2 variants and reveals a role for ORF8

Bruno A. Rodriguez-Rodriguez[1,12], Grace O. Ciabattoni[1,12], Ralf Duerr [1,2,3], Ana M. Valero-Jimenez[1], Stephen T. Yeung[1], Keaton M. Crosse[1], Austin R. Schinlever[1], Lucie Bernard-Raichon[1], Joaquin Rodriguez Galvan[1], Marisa E. McGrath [4], Sanjay Vashee [5], Yong Xue[5], Cynthia A. Loomis[6], Kamal M. Khanna [1,7], Ken Cadwell [8,9,10], Ludovic Desvignes[2,11], Matthew B. Frieman [4], Mila B. Ortigoza [1,2,13] ✉ & Meike Dittmann [1,13] ✉

Small animal models have been a challenge for the study of SARS-CoV-2 transmission, with most investigators using golden hamsters or ferrets. Mice have the advantages of low cost, wide availability, less regulatory and husbandry challenges, and the existence of a versatile reagent and genetic toolbox. However, adult mice do not robustly transmit SARS-CoV-2. Here we establish a model based on neonatal mice that allows for transmission of clinical SARS-CoV-2 isolates. We characterize tropism, respiratory tract replication and transmission of ancestral WA-1 compared to variants Alpha (B.1.1.7), Beta (B.1.351), Gamma (P.1), Delta (B.1.617.2), Omicron BA.1 and Omicron BQ.1.1. We identify inter-variant differences in timing and magnitude of infectious particle shedding from index mice, both of which shape transmission to contact mice. Furthermore, we characterize two recombinant SARS-CoV-2 lacking either the ORF6 or ORF8 host antagonists. The removal of ORF8 shifts viral replication towards the lower respiratory tract, resulting in significantly delayed and reduced transmission in our model. Our results demonstrate the potential of our neonatal mouse model to characterize viral and host determinants of SARS-CoV-2 transmission, while revealing a role for an accessory protein in this context.

Despite worldwide vaccination efforts and increasing natural immunity, emerging SARS-CoV-2 variants continue to infect and burden the health of millions of people. Past variants of concern include Alpha (B.1.1.7), Beta (B.1.351), Gamma (P.1), Delta (B.1.617.2) and Omicron (B.1.1.529), while Omicron sub-lineage XBB.1.5 currently dominates the first half of 2023[1]. Variants differ in key genes across the viral genome, including Spike (S), ORF1a, ORF1b, Nucleocapsid (N), ORF3a, ORF6, ORF7a, ORF8, ORF9b, Envelope (E), and Membrane (M). Much attention has been given to changes in Spike, as this envelope glycoprotein is the antigen targeted for most vaccination strategies to date[2] and is key to viral entry into cells[3]. Other key hotspots that accumulate mutations in SARS-CoV-2 variants are accessory proteins, of which SARS-CoV-2 encodes 8, and some serve as antagonists of the antiviral host response, most notably type I interferon production and response[4–6]. Our knowledge of SARS-CoV-2 ORFs stems from inferring functional similarities with SARS-CoV-1 and other Coronaviruses as well as functional studies using ORF cDNA overexpression constructs or full recombinant SARS-CoV-2[7–10]. The high number of SARS-CoV-2 infections coinciding with the appearance of new variants raised concerns about enhanced

transmission of these variants, posing considerable ramifications for resolution of the COVID-19 pandemic[11].

Molecular characterization of SARS-CoV-2 variants is essential to our ability to develop appropriate antiviral strategies. Previous studies have characterized SARS-CoV-2 variants by evaluating receptor binding and affinity, antigenic escape and replication dynamics as well as pathogenesis and immune evasion[12,13]. However, comparative studies on variant transmission and the molecular mechanisms governing variant-specific transmission differences are still scarce. This is partially due to limitations inherent to current animal models, such as ferrets or hamsters. While excellent models for the study of SARS-CoV-2 pathogenesis and transmission[14–16], they require special housing, the number of contact animals per index is limiting, they lack species-specific reagents, and have no or limited availability of genetic manipulation to perform mechanistic studies on host factors of transmission. In contrast, mice offer a versatile and readily available genetic toolbox, and reagents are widely available; yet, adult mice do not efficiently transmit respiratory viruses, such as influenza viruses, despite being susceptible to infection[17]. We previously overcame this hurdle for influenza virus by using neonatal mice[17], a model that has been used to study bacterial infection for over 30 years[18,19], and transmission for 7 years[17,19–22]. The model has also successfully been applied by others to study mouse parvovirus transmission[23]. Our previous study revealed influenza virus strain-specific transmission differences, the role of humoral immunity in protection of influenza virus transmission, and the influence of sialidase expression during influenza virus–*Streptococcus pneumonia* co-infection[17]. However, the model is not established for SARS-CoV-2.

Here, we present a neonate K18-hACE2 mouse model that allows for transmission of SARS-CoV-2 between pups of the same litter, and we examine side-by-side the transmission of early and current SARS-CoV-2 variants of concern (VOC). In addition, we characterize the transmission of two recombinant SARS-CoV-2, lacking accessory proteins ORF6 or ORF8. Our study highlights the power of our model to elucidate the dynamics of SARS-CoV-2 variant transmission and provides evidence of an accessory protein, ORF8, to be critical for SARS-CoV-2 transmission in neonatal mice. Our tractable small animal model will help decipher some of the most critical factors involved in the transmission of SARS-CoV-2.

## Results

### Neonatal K18-hACE2 mice efficiently support SARS-CoV-2 WA-1 transmission

While influenza viruses readily infect wild-type mice, the ancestral SARS-CoV-2 and the previously clinically important delta variant rely on the human version of the SARS-CoV-2 host cell receptor, ACE2, and cannot engage murine Ace2[24]. In contrast, a number of other SARS-CoV-2 variants acquired specific Spike mutations, most notably N501Y, which allows them to bind murine Ace2[25]. Ultimately, we aimed to compare a panel of variants, from Alpha to Omicron, to the ancestral SARS-CoV-2. With this in mind, we utilized K18-hACE2 mice in our study, which express human ACE2 under the control of the K18 promoter[26].

For SARS-CoV-2, data on transmission in adult mice is scarce, with one study reporting SARS-CoV-2 B.1.351 (Beta) transmission from index mice infected with 5E6 FFU via close contact[27]. Transmission between co-housed mice in the same cage, determined by the presence of viral RNA in or seroconversion of contacts, was 41 and 8%, respectively. Infectious virus was not assessed. Transmission of influenza viruses in adult mice has also been inefficient and inconsistent with transmission events being difficult to reproduce between research groups[17,28,29]. This rendered the adult mouse model an unreliable tool to study transmission of influenza viruses. To determine the efficiency of SARS-CoV-2 transmission in adult mice in our hands, we infected 13-week-old K18-hACE2 index mice intranasally, under

anesthesia, with a lethal dose (10,000 PFU titered in VeroE6-TMPRESS2-T2A-ACE2 cells) of ancestral SARS-CoV-2 USA_WA-1/2020 (WA-1). Starting at the day of infection, we co-housed the infected index mice with naïve contacts at a ratio of 1 index to 2–3 contacts (Supplemental Fig. 1a). We monitored morbidity (weight loss) and mortality (humane endpoint) and collected longitudinal nasal shedding samples non-invasively and without anesthesia, by dipping the nares in viral media. Thus, we were able to leverage the kinetics of upper respiratory tract (URT) shedding in individual mice, longitudinally, as a measure of viral infection. In contrast to index mice, we did not observe morbidity or mortality in contact mice by the end of the experiment (10 days postindex infection) (Supplemental Fig. 1b, c). Index mice shed infectious particles starting from day 1-6 postinfection, with peak viral shedding at day 2 (Supplemental Fig. 1d). We detected infectious virus in URT shedding samples at any time point in only 1/9 (11% total) of the contact animals (Supplemental Fig. 1d). This was the same contact animal that still had infectious titers in the URT and the lung at 10 dpi (Supplemental Fig. 1e). By cage, this represented 0/3 (0%), 0/3 (0%), or 1/3 (33%) transmission efficiency. In a parallel experiment, we determined transmission by contact seroconversion (Supplemental Fig. 1f). PBS-infected mice served as negative control, and mice infected at a sublethal dose (1,000 PFU) as a positive control for seroconversion. At the experimental endpoint of 22 days postindex infection, we found that 1/5 (20%) of contact mice had seroconverted (Supplemental Fig. 1f). By cage, this represented 0/3 (0%), or 1/2 (50%) transmission. We concluded that contact transmission of WA-1 in adult K18-hACE2 mice does occur, albeit at low efficiency and with high cage-to-cage variability. Thus, like the influenza virus model of transmission, we next set out to establish SARS-CoV-2 infection in neonate mice.

We first performed a dose-response experiment to determine the minimum viral dose required to yield robust SARS-CoV-2 infection and shedding in neonates. We combined C57BL/6 K18-hACE2[+/+] males and C57BL/6 (hACE2[-/-]) females to produce SARS-CoV-2 WA-1 permissive K18-hACE2[+/-] offspring. At 4-7 days of age, pups were infected intranasally with 3 μL of ancestral WA-1 without anesthesia. We used an escalating viral dose of 1,500, 15,000 or 50,000 PFU. We then monitored morbidity (lack of weight gain) and mortality (humane endpoint) and collected daily longitudinal nasal shedding samples by dipping the nares in collection media (Supplemental Fig. 1g). Pups infected with 1,500 PFU failed to gain further weight by day 3 postinfection (3 dpi), and they all succumbed to infection by 4 dpi. Pups infected with 15,000 or 50,000 PFU died at 3 dpi before detectable weight loss occurred (Supplemental Fig. 1h, i). Nasal shedding titers between pups infected with 1,500, 15,000, or 50,000 PFU were similar on 1 and 2 dpi, but became apparent on 3 dpi (Supplemental Fig. 1j): titers from pups infected with 1,500 PFU declined compared to 2 dpi, whereas titers from pups infected with 15,000 PFU remained at the same level as 2 dpi, and titers from pups infected with 50,000 PFU increased beyond that detected at 2 dpi. The detected shedding titer difference between the 1,500 and 50,000 PFU groups at 3 dpi was about 100-fold. This shows that an escalating input titer can modulate the dynamic of viral shedding in our model. As the lowest tested dose of 1,500 PFU yielded robust infection while leaving an additional day for transmission to occur before index mice succumb to infection, we used 1,500 PFU as the standardized inoculum in subsequent neonatal mouse infection experiments.

Next, we tested intra-litter transmission of WA-1 in neonatal mice. We infected 4–7-day old K18-hACE2[+/-] index mice with 1,500 PFU of WA-1 and placed them back with the (nonpermissive) dam and their (permissive) naïve littermates at a ratio of 1 index to 4-6 contact mice (Fig. 1a). We first observed that both morbidity and mortality were offset in contact mice by 2-3 days compared to index mice (Fig. 1b, c), indicating successful transmission. In index mice, we detected SARS-CoV-2 RNA as early as 1 dpi, peaking at 2 dpi, whereas some contact

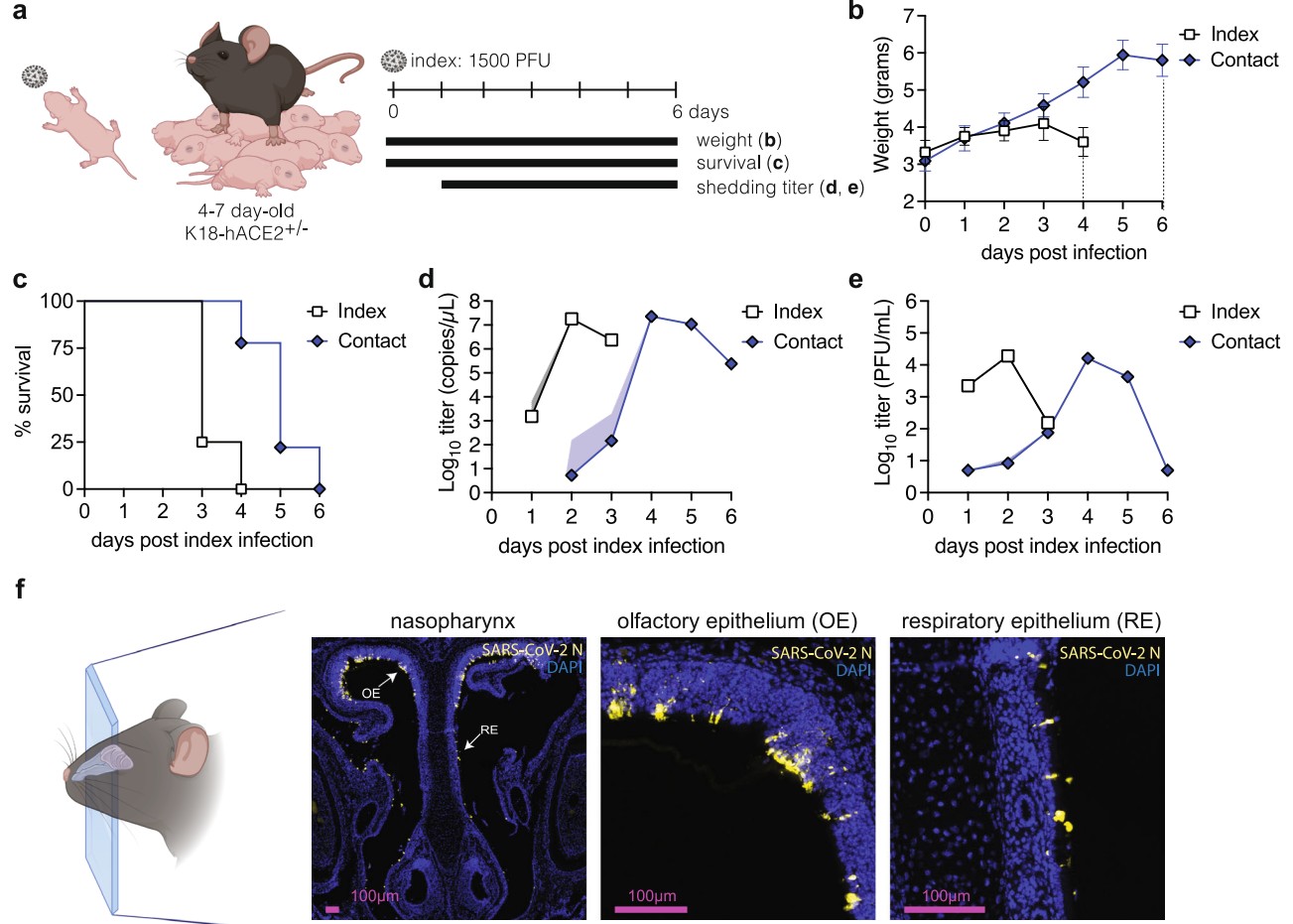

**Fig. 1 | Establishment of a neonatal K18-hACE2 mouse transmission model using SARS-CoV-2 WA-1. a** Four to seven-day-old K18-hACE2[+/-] pups were intranasally infected with 1500 PFU of SARS-CoV-2 WA-1 and cohoused with uninfected littermates for 6 days. Weight and survival were monitored daily, and viral shedding samples were collected by dipping the nares of each individual pup in viral medium daily. Data from two independent repetitions, with a total of $n = 4$ index and $n = 9$ contact mice. Mean body weight (**b**) and survival (**c**) of index and contact pups. Viral burden in shedding samples analyzed by RT-qPCR for SARS-CoV-2 RNA (**d**) and by plaque assay for infectious virus (**e**). Data shown as geometric mean (line) with geometric standard deviation (shaded area). Individual values below the limit of detection (50 PFU/ml) were set to 5. **f** Immunohistochemistry for SARS-CoV-2 N protein in 4-7 day old mice nasopharynx. Pups were infected intranasally with 1500 PFU of SARS-CoV-2 WA-1, and heads were fixed at 2 dpi, paraffin embedded, sectioned through the nasopharynx, and stained for SARS-CoV-2 N protein (yellow) and DAPI (blue) for nuclei. Arrows represent areas magnified in the adjacent panels. OE and RE indicate olfactory epithelium and respiratory epithelium, respectively. Created with BioRender.com.

pups began shedding viral RNA from 2 dpi, peaking at 4 dpi (Fig. 1d). A parallel experiment with heat-inactivated SARS-CoV-2 showed that the viral genomes in shedding samples were due to active viral replication and not carryover from inocula (Supplemental Fig. 1k). Detecting SARS-CoV-2 RNA in contact mice was thus another indication of transmission. However, we stringently define transmission as the sustained detection of infectious viral particles in contact mice. Therefore, we determined viral titers in nasal shedding samples by plaque assay. Infectious particles from index mice (Fig. 1e) correlated with the onset and waning of RNA detection (Fig. 1d). Detection of infectious particles in contact pups confirmed SARS-CoV-2 transmission. Infectious particles in contacts peaked on day 4 and decreased by days 5 and 6, similar to SARS-CoV-2 RNA trends. Of note, all contact pups shed infectious particles by day 4, representing 16/16 (100%) transmission of WA-1 in our model.

Detection of infectious particles in shedding samples of K18-hACE2 neonatal mice suggested robust viral infection in the URT. To determine the sites of infection within the URT in spatial granularity, we performed immunohistochemistry (IHC) on the nasopharynx of index pups. Heads of neonatal mice were harvested at their peak of viral shedding (2 dpi), sectioned through the nasopharynx, and stained

for SARS-CoV-2 N protein. We detected SARS-CoV-2-positive cells in the upper olfactory and respiratory epithelium lining, demonstrating replication of SARS-CoV-2 WA-1 in cells of the URT (Fig. 1f). Taken together, these results establish and validate a tractable neonatal K18-hACE2 mouse model for the transmission of SARS-CoV-2.

## SARS-CoV-2 variants of concern display distinct replication dynamics and tropism in neonatal index mice

High transmissibility of SARS-CoV-2 Alpha and Delta variants has been proposed to be driven by efficient URT replication and shedding[30,31]. This is in contrast to the influenza virus literature, where URT replication is not correlated with aerosol generation or transmission in humans[32]. Using the neonate mouse model, we experimentally recapitulated that influenza virus transmission efficiency does not correlate with URT titers, but rather with the amount of exhaled virus in shed secretions[17]. Others have shown that influenza virus transmission is dependent on both the timing to and magnitude of peak shedding[33,34]. Similar observations were noted for SARS-CoV-2 in Syrian hamsters where oral virus titers from swabs was a poor proxy for airborne shedding, yet peak airborne virus shed into the environment correlated with transmission[35].

To identify the index correlates of SARS-CoV-2 transmission in our model, we characterized viral loads in various respiratory compartments: lower respiratory tract (lung homogenates), URT (retro tracheal lavages), or shed secretions (expelled virus). We began by evaluating the dynamics of viral replication within these compartments for the ancestral WA-1 virus. 4–7-day old K18-hACE2$^{+/-}$ pups were infected with 1,500 PFU of WA-1 and measured infectious particle load in the respective sample type at 1, 2 and 3 dpi (Fig. 2a, b, Supplemental Fig. 2a). At 1 dpi, expelled and URT titers were both higher than lung titers, albeit not statistically significant. At 2 dpi, expelled, URT, and lung titers were similar in titer to each other. At 3 dpi, shedding titers declined, while lung titers increased, resulting in lung titers that were significantly higher than shedding titers. This suggests that in our model, SARS-CoV-2 WA-1 infection administered in low volume intranasally without anesthesia progresses from the URT to the lungs over time.

Next, we compared URT shedding dynamics of ancestral virus and variants. We first generated clonal (plaque-purified), sucrose-purified, and deep-sequenced viral working stocks. This rigorous quality-control pipeline ensures the identity of VOCs (i.e., the presence of variant-defining mutations) and their integrity (the lack of tissue culture-induced mutations). We then infected K18-hACE2$^{+/-}$ pups with 1,500 PFU of either Alpha (B.1.1.7), Beta (B.1.1.351), Gamma (P.1), Delta (B.1.617.2), Omicron BA.1, or Omicron BQ.1.1. We first determined the dynamics of infectious virus shedding for each isolate from 1 to 4 dpi or until the pups succumbed to infection (Fig. 2a, Supplemental Fig. 3). Shedding from mice infected with ancestral WA-1 peaked at 2 dpi before decreasing by 3 dpi (Fig. 2b). In contrast to WA-1, shedding from mice infected with Alpha peaked earlier, at 1 dpi, and steeply decreased by 2 dpi (Fig. 2c). Shedding from mice infected with Beta also peaked at 1, but, in contrast to Alpha, remained at the same level on 2 dpi before declining on 3 dpi, resulting in a titer similar to WA-1-infected pups (Fig. 2d). Shedding from mice infected with Gamma, and Delta had a slope similar to WA-1, with a peak at 2 dpi, before decreasing by 3 dpi (Fig. 2e, f). Shedding titers from Gamma-infected pups were on average lower than those from WA-1-infected pups, albeit not statistically significant (Fig. 2e). Mice infected with Omicron isolates BA.1 or BQ1.1 shed overall low levels of virus, with most replicates below the detection limit (Fig. 2g, Supplemental Fig. 2d). However, these two Omicron variants differ with respect to shedding kinetics: BQ.1.1 shedding peaked at 1 dpi versus BA.1 at 3 dpi (Fig. 2g, Supplemental Fig. 2d). Thus, our model is able to capture the unique kinetics of SARS-CoV-2 variant URT shedding from infected pups.

We next compared the sites of viral replication for each variant and found the distribution of viral load in different sample types to be distinct for specific viruses. For Alpha, shedding and URT titers at 2 dpi were comparable in magnitude to 2 dpi WA-1 (Fig. 2b, c), but lung titers were trending lower (Fig. 2c). To determine whether these lower lung titers were a consequence of Alpha's earlier onset and waning of replication and shedding, we assayed Alpha viral loads in the three compartments also at Alpha's peak day of shedding, 1 dpi (Supplemental Fig. 2b, left panel). Again, we found significantly higher titers in the shedding and URT samples than in the lungs, suggesting that, in our model, both on 1 and 2 dpi, Alpha replicates better in the URT. For Beta, Gamma, and Delta we found no significant differences in titers between shed virus, URT, and lungs, suggesting equal ability to infect and replicate in both the URT and the lower respiratory tract and to be shed from the URT (Fig. 2d-f). Indeed, IHC of the URT region confirmed that WA-1, Alpha, Beta, Gamma, and Delta infect the URT efficiently (Fig. 2h). In contrast, BQ1.1 titers in URT and lungs at 2 dpi were undetectable (Fig. 2g), and IHC did not reveal Omicron BQ.1.1-infected cells in the URT epithelium (Fig. 2h). To determine whether these undetectable titers were a consequence of BQ1.1's earlier onset and waning of replication and shedding, we assayed Omicron BQ.1.1 viral loads in the three compartments also at its peak day of shedding, 1 dpi.

We found that average titers were still low in all three compartments, and that some samples were above the limit of detection in shedding and URT samples (Supplemental Fig. 2c). These results were similar to 1 dpi data obtained for Omicron BA.1 (Supplemental Fig. 2e, f). No infectious Omicron particles were detected in the lungs for either Omicron sub-variant, suggesting that Omicron does not replicate efficiently in neonatal K18-hACE2 mice (Fig. 2g, Supplemental Fig. 2e). Our results are in line with findings from others, showing that Omicron variants are attenuated in hamsters and mice, despite the presence of its receptor hACE2 in the K18 model[24,36,37].

Interestingly, viral loads in shedding samples were similar to those from their respective retro tracheal lavage samples for all viral isolates, showing that, in our model, SARS-CoV-2 viruses assayed do not display a defect in viral expulsion, and thus, URT shedding can be used as a proxy for SARS-CoV-2 URT replication.

Taken together, our results demonstrate that, except for Omicron variants, our model allows for efficient replication and shedding of SARS-CoV-2 variants at comparable levels to the ancestral WA-1. Temporal shedding kinetics differ between variants and permits this model to be used as a tool to understand the mechanisms of variant SARS-CoV-2 transmission.

## Neonatal mouse model reveals dynamics in SARS-CoV-2 variant transmission

We next utilized our panel of SARS-CoV-2 variants ranging from early to current pandemic isolates to test transmission. 4-7-day-old index K18-hACE2$^{+/-}$ mice were infected with either ancestral SARS-CoV-2 WA-1, Alpha, Beta, Gamma, Delta, Omicron BA.1 or Omicron BQ.1.1 and cohoused with naïve neonatal mice at a 1:6-9 ratio. WA-1, Alpha-, Beta-, and Delta-infected index mice all succumbed to infection by 3 dpi (Supplemental Fig. 3a-c, e). Gamma-infected index mice succumbed by 4 dpi (Supplemental Fig. 3d). For Delta- and Omicron BA.1-infected index mice, morbidity (lack of weight gain) and mortality were delayed, and for Omicron BQ.1.1-infected index mice, we did not detect substantial morbidity but complete, yet delayed, mortality of index mice (Supplemental Fig. 3e-g). Transmission events occurring at late time points, after the death of index pups, may stem either from late onset of contact shedding or from sequential transmission between contacts. Except for the two Omicron variants, some, or all contact mice in cages with other variants succumbed to infection by 7 dpi (Supplemental Fig. 3).

We next investigated the dynamics of transmission using two non-redundant readouts from contact pups: 1. Timing of viral acquisition by contacts: We quantify virus acquisition by contact pups as "transmission events" when virus is detected at least twice in shedding samples (Fig. 3). The first day of viral shedding from each contact ("infectious virus-positive") is counted as the onset of acquisition. This lets us evaluate viral acquisition as a function of time and compare the kinetics of acquisition between variants. This readout does, however, not depict the amplitude of contact infection. 2. Detection of viral titers from URT shedding of contact animals: This measure allows the detection of subtle variations in the level of contact shedding. The differences in shedding amplitude observed in our experiments may stem from the variations in shedding by the index mice (i.e., infectious dose and timing of transmission by the index), and/or differences in the viral replicative fitness in contacts. Changes in contact shedding titers may matter in sequential transmission experiments, where contacts become index for naïve individuals. For WA-1 and Alpha, 100 % transmission was reached by 3 or 4 dpi, respectively, with slopes that were not significantly different (Fig. 3a, left panel). Both ancestral WA-1 and Alpha infectious particles were detected in some contact mice as early as 1 dpi, and Alpha's were significantly higher than WA-1's on 3 dpi, suggesting a slight transmission advantage by Alpha. Alpha contact shedding titers peaked at 4 dpi, and faded earlier than WA-1, starting at 4 dpi, (Fig. 3a, right panel), which is in line with the early drop of shedding titers previously observed in Alpha index mice

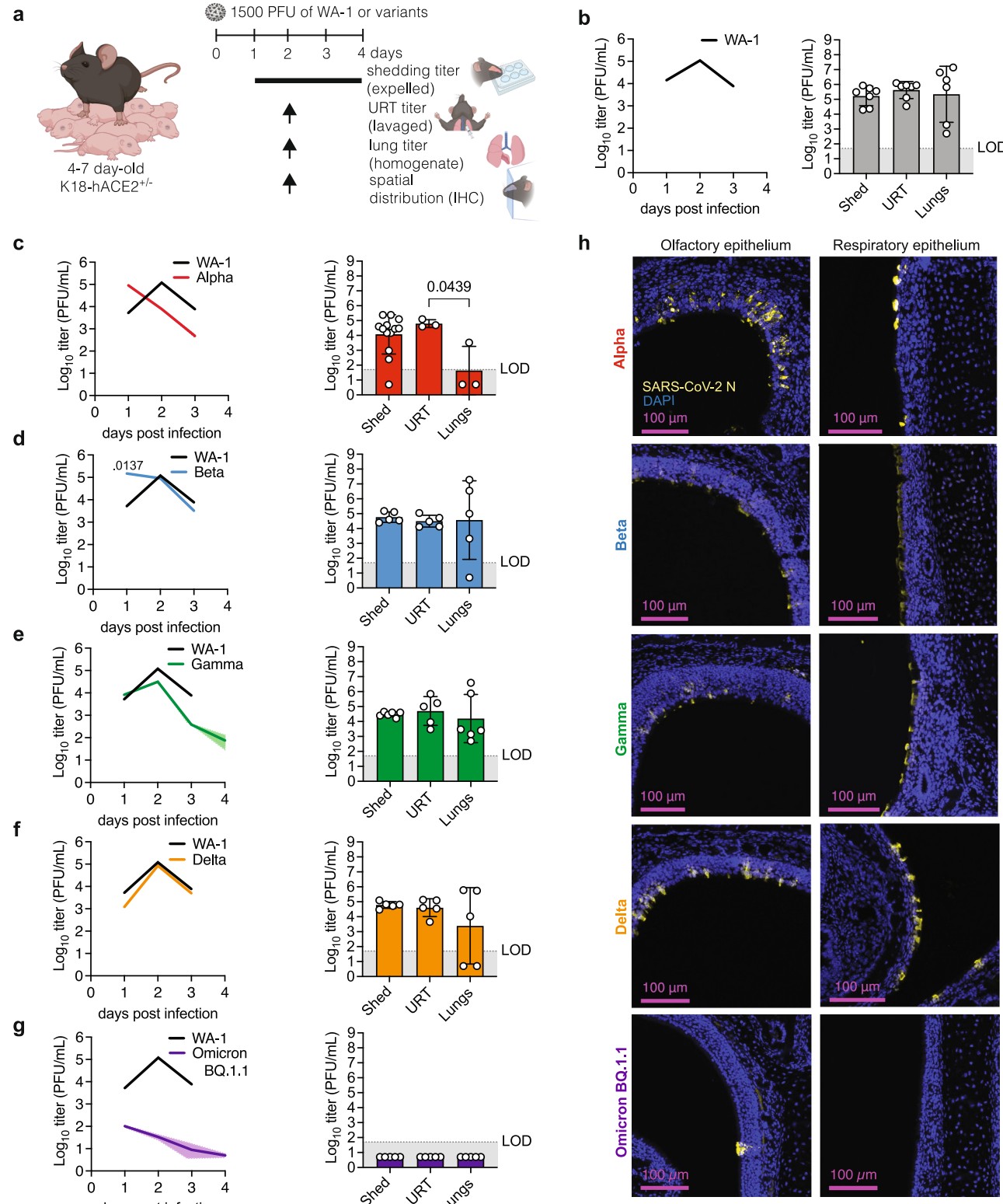

**Fig. 2 | SARS-CoV-2 variant replication dynamics and tropism in index mice.**
**a** Neonatal K18-hACE2⁺/⁻ mice were infected with indicated SARS-CoV-2 and viral shedding samples were collected daily. At 2 dpi, retrotracheal lavages and lungs were collected to determine viral titers and heads were fixed for immunohistochemistry. **b**–**g** Viral burden in daily shedding samples (left) and at 2 dpi in shedding samples, upper respiratory tract and lungs (right). Individual values below the limit of detection (LOD, 50 PFU/ml) were set to 5. Data from at least 2 independent repetitions with $n$ = 6 - 15 pups per group. Only significant p-values (Kruskal-Wallis test) are presented. **h** Immunohistochemistry for SARS-CoV-2 N at 2 dpi in nasopharynx. Created with BioRender.com.

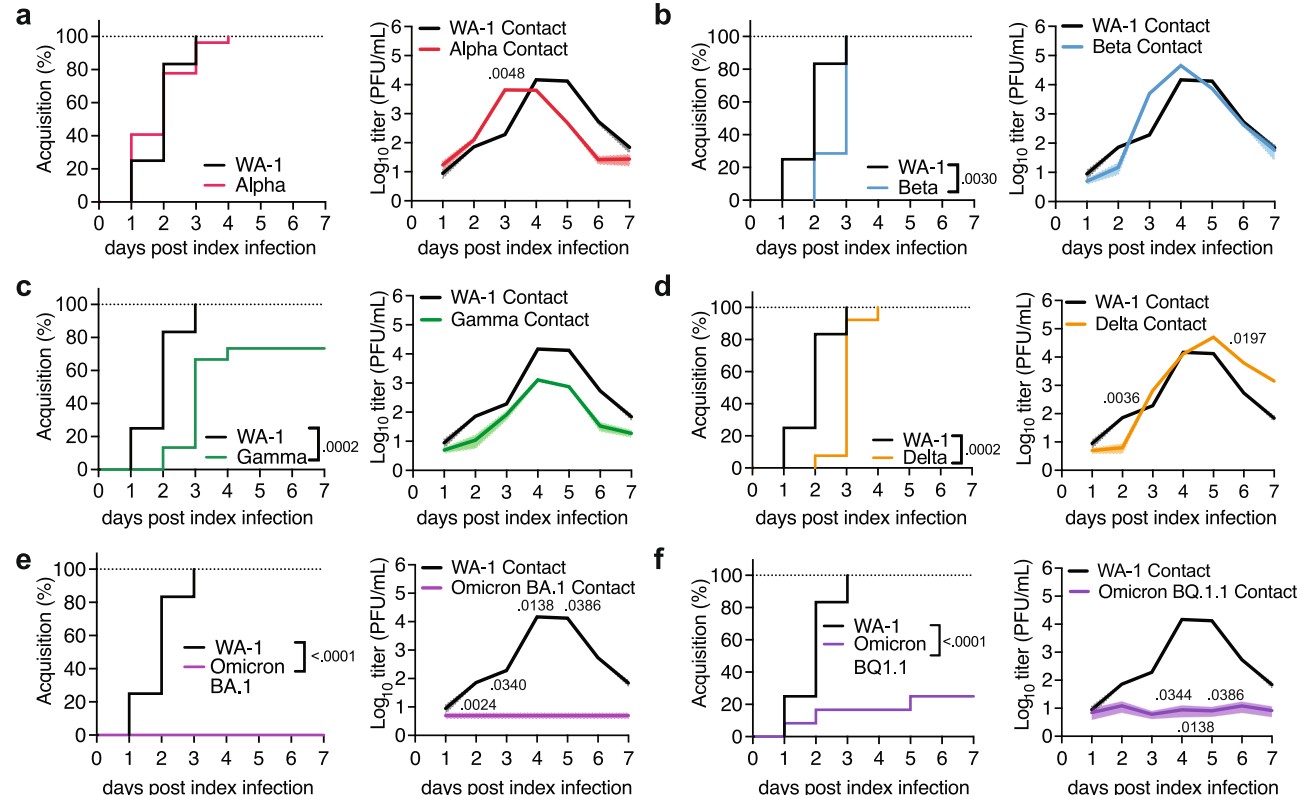

**Fig. 3 | SARS-CoV-2 variant transmission in neonatal mice.** Neonatal K18-hACE2[+/-] mice were infected with indicated SARS-CoV-2 and co-housed with uninfected littermates for 7 days at a 1:6-9 ratio. The acquisition of infection by and viral burden in contact mice was monitored daily. Ancestral WA-1 was compared to Alpha (**a**), Beta (**b**), Gamma (**c**), Delta (**d**), Omicron BA.1 (**e**), and Omicron BQ1.1 (**f**) variants. Left panels show the percentage of viral acquisition in inverted Kaplan-Meier plots. The onset of acquisition was scored as the first day of sustained infectious virus detection. Right panels show the viral burden in contact shedding samples determined by plaque assay. Data is shown as geometric mean (line) with geometric standard deviation (shaded area). Individual values below the limit of detection (50 PFU/ml) were set to 5. Data from at least 2 independent repetitions $n = 1$ index and 4-6 contact pups per repetition. Only significant p-values (Mantel-Cox Log-rank test for Kaplan-Meier plots, Kruskal-Wallis test for viral burden) are presented.

(Fig. 2c). In contrast to Alpha, Beta contact mice displayed significantly delayed acquisition compared WA-1 (Fig. 3b, left panel), but reached similar peak contact titers (Fig. 3b, right panel). Gamma contact mice showed significantly delayed acquisition compared to WA-1 (Fig. 3c, left panel), and achieved a 73% transmission overall. Gamma contact mice also had an overall lower viral load, although kinetics to and from peak titer was similar to that of WA-1 (Fig. 3c, right panel). Like Gamma, Delta contact mice significantly lagged WA-1 contact mice regarding acquisition (Fig. 3d, left panel), and Delta contacts peak infectious virus levels were similar to WA-1 (Fig. 3d, right panel). Delta-infected contacts continued shedding infectious particles until the end of the experiment at 7 dpi (Fig. 3d, right panel). Omicron BA.1 contact mice did not acquire infection in our model or shed infectious virus (Fig. 3e), which is in line with other reports showing that Omicron BA.1 is attenuated, and airborne transmission is reduced in rodents[24,36–38]. Interestingly, and in contrast to BA.1, Omicron BQ.1.1 contact mice achieved 25% transmission in our model (Fig. 3f, left panel). BQ.1.1. index shed with exceptionally low viral titers (Fig. 2g) which corresponded with a lower transmission rate and low contact shedding (Fig. 3f, right panel). The transmission difference between BA.1 and BQ.1.1 was surprising given that peak shedding titers of both Omicron subvariants in index animals were equally low (Fig. 2g, Supplemental Fig. 2c). It is possible that the timing of peak shedding (1 dpi for BQ.1.1 vs 3 dpi for BA.1) is the critical determinant for Omicron transmission in our model. Overall, our data shows that the magnitude of index shedding corresponds with the success of transmission to contacts (Supplemental Fig. 2g).

Next, we characterized the URT inflammatory repertoire of index mice after viral challenge in our model. An antiviral/inflammatory

response to infection is expected to attenuate the virus within the host, causing reduced transmission[5,39]. However, increased inflammation may also induce URT secretions that help expel the virus into the environment, which may favor transmission[20,22]. We analyzed the cytokines present in URT shedding samples of index mice by multiplex ELISA at 6, 24 and 48 h, using heat-inactivated (HI) SARS-CoV-2 WA-1 or poly (I:C) as controls. We detected the presence of multiple cytokines in poly(I:C)-treated mice at the 48 h timepoint, albeit at levels that were lower than in WA-1-infected mice (Supplemental Fig. 4a). HI WA-1 failed to induce measurable inflammation across timepoints, whereas we detected increased cytokine levels in samples from WA-1-infected mice by 48 h (Supplemental Fig. 4a). This suggests that our purified viral stocks do not contain exogenous inflammatory material, that active WA-1 replication is required to drive inflammation in our model, and that the 48 h timepoint is optimal to measure the cytokine profile in URT shedding samples.

To explore a potential association between index URT inflammation and transmission, we measured cytokines in shedding samples of index mice infected with either ancestral SARS-CoV-2 WA-1 or variants Alpha, Beta, Gamma, Delta, Omicron BA.1 and Omicron BQ.1.1 at 48 hpi. Comparable infectious viral shedding titers at this time point for most variants, between $10^4$ and $10^5$ PFU/mL (Fig. 2), allowed for qualitative comparison of cytokine signatures between these equally replicating viruses. Both Omicron variants had much lower viral shedding titers at 48 hpi ($10^1$ PFU/mL, Fig. 2g), and accordingly, the cytokine signatures for both Omicron variants at 48 hpi were quiet (Supplemental Fig. 4b). Notably, at 24 hpi, which is the peak time of shedding for BQ.1.1, we detected increased cytokine levels in

BQ.1.1-infected index (Fig. 2g, Extended Data 4c), but found no such increase in BA.1-infected mice, where shedding peaks later, at 72 hpi (Supplemental Fig. 2d, Supplemental Fig. 4c). For equally replicating viruses (WA-1, Alpha, Beta, Gamma, and Delta) we found that the cytokine signature upon ancestral WA-1 infection was the most different from signatures upon infection with variants (Supplemental Fig. 4b). Gamma and Alpha signatures clustered together, and so did Alpha and Delta signatures, respectively. The set of key upregulated cytokines was similar between equally replicating viruses (Supplemental Fig. 4b, upper left quadrant), but the amplitude of upregulation differed between viruses. Notably, cytokine cluster distance by the different viruses correlated with their respective transmission efficiency (Supplemental Fig. 4d), except Alpha. Further studies are needed to decipher whether any and if yes, which specific URT cytokines induced by index pups contribute to transmission (Supplemental Fig. 4b and Fig. 3). Our model and the availability of transgenic mice with interruptions in key cytokine pathways uniquely enables us to perform these future studies.

Taken together, our results show that our neonatal mouse model can characterize differences in transmission dynamics inherent to SARS-CoV-2 variants. Our daily method of viral sampling enables tracking viral replication in individual index and contact mice over time, providing granularity in transmission parameters, including the association of variant-specific cytokine signatures and transmission.

## Accessory proteins ORF6 and ORF8 play a role in successful SARS-CoV-2 transmission

In addition to mutations in spike, several SARS-CoV-2 variants display mutations in accessory genes, such as *ORF3*, *ORF6*, *ORF7*, and *ORF8*, which have been implicated in the interference of immune signaling or direct host effector counteraction[6,8,10,40–44]. The selective advantage of changes in these accessory proteins, if any, remains undefined, particularly regarding transmission. We thus aimed to determine how the lack of specific accessory proteins impacts SARS-CoV-2 transmission in our model, and focused on two accessory proteins: ORF8, because it has mutations defining the Alpha, Gamma, and Delta variants, and, ORF6, which lacks mutations in the variants utilized in this study but is mutated in some Omicron variants (i.e. BA.2, BA.4), in addition to being one of the best-characterized SARS-CoV-2 accessory proteins[1,42–44]. Both of these recombinant viruses have been characterized in cultured cells and in adult K18-hACE2 mice[45], revealing reduced viral titers in vitro, but similar lung titers in vivo. Interestingly, the virus lacking ORF8 displayed increased lung pathology score[45], which may suggest an increased inflammatory process is induced by this virus.

We infected K18-hACE2$^{+/-}$ neonatal mice with 1500 PFU of recombinant SARS-CoV-2 WA-1 (rWA-1), recombinant SARS-CoV-2 lacking ORF6 (ΔORF6), or recombinant SARS-CoV-2 lacking ORF8 (ΔORF8). Similar to our previous observations with the clinical WA-1 isolate (Fig. 2b), shedding titers from rWA-1 index mice peaked at 2 dpi and decreased by 3 dpi (Fig. 4a, left panel). Index shedding from ΔORF6 infected mice was similar to rWA-1 in kinetics, with a trend of lower magnitude (5-fold at 2 dpi, Fig. 4b, left panel). Index mice infected with ΔORF8 consistently shed up to 100-fold less virus than mice infected with rWA-1 (Fig. 4c, left panel) and survived 1 day longer than rWA-1 infected index (Supplemental Fig. 5a-c), hence still shedding at 4 dpi. Next, we analyzed virus replication in shedding samples (expelled virus), URT lavages (URT replication), and lungs (lower respiratory tract replication) of index mice. Like the WA-1 isolate, we observed much higher rWA-1 titers in lungs than in the URT and shedding samples (Fig. 4a, right panel). For ΔORF6, we obtained similar titers in all three sample types, 1×10$^4$ PFU/mL on average, although there was a broad distribution with some pups displaying higher viral titers in the lung than in shedding or URT samples (Fig. 4b, right panel). We concluded that, in our model, the lack of ORF6 did not significantly attenuate viral replication, shedding, or significantly

change tissue tropism with respect to parental rWA-1. This was different for ORF8. ΔORF8 shedding and URT titers were lower than for rWA-1 or ΔORF6 (Fig. 4c, left panel). Surprisingly, viral burden in the lungs was comparable to those of rWA-1 and WA-1 ΔORF6. This was similar to previous findings in adult mice, where the lack of ORF8 did not alter lung titers[41,45]. Thus, the lack of ORF8 seems to significantly reduce ΔORF8's ability to robustly replicate specifically in the URT.

We next performed transmission experiments with the three recombinant viruses. Like the WA-1 isolate, rWA-1 acquisition by contacts occurred at 1 dpi, was complete by 4 dpi, and contact shedding titers decreased from 4 dpi onwards (Fig. 4d). ΔORF6 was similar to parental rWA-1 in both contact shedding onset, dynamics, and acquisition (Fig. 4e). Together, these results suggest that the lack of ORF6 does not significantly attenuate transmission in our model. Onset of acquisition for ΔORF8 was similar to rWA-1 and ΔORF6, 1 dpi, but in contrast to the other two recombinant viruses, the slope of acquisition was slower, and transmission was 76% (Fig. 4f). We hypothesize that this reduction in transmission efficiency is a consequence of the lower shedding titers by the index mice.

Next, we analyzed cytokine levels in URT and lung samples from index mice by multiplex ELISA (Fig. 4g). We found that cytokine levels present in the URT of mice infected with ΔORF8 had a similar magnitude to those infected with rWA-1 and ΔORF6, despite 100-fold lower WA-1 ΔORF8 titers at the same time point (2 dpi). This suggests increased inflammatory signature of ΔORF8 relative to amount of virus present in the URT. In lungs, which display similar viral titers across the three viruses (Fig. 4g), we observed a similar pattern of cytokines between ΔORF8- and rWA-1-infected mice (Fig. 4g). How and whether inflammation by ΔORF8 attenuates viral replication specifically in the URT, but not in the lungs, remains the topic of further investigation. Of note, cytokine levels were elevated in ΔORF6-infected lungs compared to both rWA-1- and ΔORF8-infected lungs, particularly IL-6, suggesting that the recombinant virus lacking ORF6 is unable to suppress production of certain cytokines in the lungs of neonatal mice.

Taken together our results show how removal of one SARS-CoV-2 accessory protein, ORF8, reduces URT replication, resulting in reduced shedding and transmission in neonatal mice. Our neonatal mouse model allows for a unique view on these molecular processes in spatial granularity, and the availability of mouse-specific reagents will enable future mechanistic studies on the role of ORF8 in URT replication and transmission.

## Discussion

Mouse models of SARS-CoV-2 infection have been utilized extensively to study the pathogenesis of emerging SARS-CoV-2 variants[46–52]. However, adult mice do not robustly transmit SARS-CoV-2 (Supplemental Fig. 1a–f and ref. 27). In our study, we developed and validated a neonatal mouse model to characterize the transmission of SARS-CoV-2 human isolates based on our previous experience with influenza A virus (Fig. 1 and ref. 17). Using a unique SARS-CoV-2 panel spanning early to current pandemic human isolates, we demonstrate how variant-inherent mutations affect tropism and shedding (Fig. 2), transmission (Fig. 3), and upper respiratory tract cytokine repertoires (Supplemental Fig. 4). Finally, our study reveals a previously unappreciated role for the accessory protein ORF8 in viral upper respiratory tract replication, inflammation, and transmission in neonatal mice (Fig. 4), displaying the power of our model to study the viral and host molecular determinants of SARS-CoV-2 transmission in mice.

Our model has several advantages over existing animal models of SARS-CoV-2 transmission: First, amenability to a wide range of available knockout animals allows for future systematic studies determining host factors that are critical for transmission. Second, our model allows for powered studies, as high numbers of contact neonatal mice are more easily achieved than in other animal models. Third, there are fewer husbandry challenges to work with mice as compared to

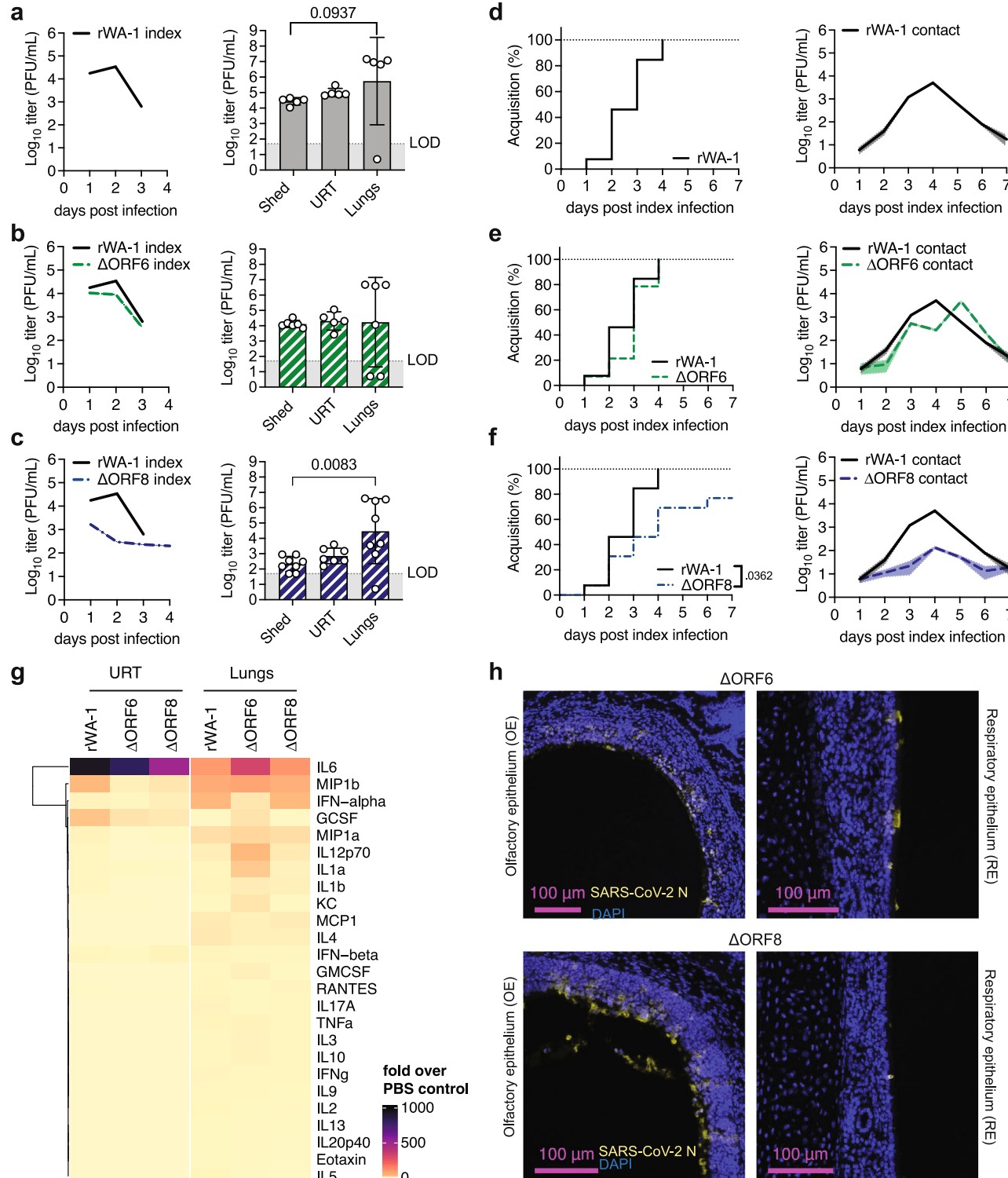

**Fig. 4 | Recombinant SARS-CoV-2 ORF6 and ORF8 replication dynamics, tropism, transmission, and cytokine profiles.** Neonatal K18-hACE2[+/-] mice were intranasally infected with 1500 PFU of recombinant WA-1 (rWA-1), rWA-1 lacking ORF6 (ΔORF6), or rWA-1 lacking ORF8 (ΔORF8). **a-c** Viral burden in daily shedding samples (left) and in 2 dpi shedding samples, upper respiratory tract (URT) and lungs (right). Data from at least $n = 5$ pups per group. **d-f** Percentage of viral acquisition shown in inverted Kaplan-Meier plots and viral infectious burden in contact shedding samples, shown as geometric mean (line) with geometric standard deviation (shaded area). Data from at least 2 independent repetitions with $n = 1$ index and 4-6 contacts each. **a-f** Only significant p-values (Mantel-Cox Log-rank test for Kaplan-Meier plots, Kruskal-Wallis test for viral burden) are presented. Individual values below the limit of detection (LOD, 50 PFU/ml) were set to 5. **g** Heatmap representing cytokine levels at 2 dpi in retrotracheal lavages and lungs measured by multiplex ELISA. Data represent -fold induction over PBS-inoculated pups. At least $n = 3$ pups per condition. **h** Immunohistochemistry for SARS-CoV-2 N at 2 dpi in nasopharynx.

hamsters or ferrets. Further, the availability of mouse-specific reagents and tools increases the potential to study virus-induced immune pathways in the host, as well as respiratory mechanics that could impact transmission, such as air flow and lung capacity. Finally, we can track the index's progression and contact infection longitudinally, due to non-invasive sampling. This provides more granularity and insight into the kinetics of viral shedding and URT viral load at the level of individuals, which illuminates our understanding of transmission efficiency.

Although these features establish our model as a unique and workable tool for studying SARS-CoV-2 transmission, other animal models, such as hamsters and ferrets, have certain advantages over the limitations of our system. One limitation of our model is that the modes of transmission cannot be tested experimentally, because suckling mice cannot be separated from each other or their mother. It is likely that transmission in the model, like in humans[53,54], occurs via a combination of modes. Long-range aerosol transmission, as can occur in humans[55], cannot be measured in our current experimental setup. Nevertheless, the model features behavioral situations favorable for both influenza virus and SARS-CoV-2 transmission, as shown both in humans and in more classical animal model systems: prolonged time of exposure[56], proximity of individuals in enclosed spaces[57] and cohabitation with household members[54,58–60]. In addition, our non-invasive sample collection method from individual pups allows us to longitudinally measure temporal dynamics and amplitude of viral shedding, both factors that are critical in respiratory virus transmission[17,33,34,61–63]. Another limitation of our model is that the expression of the hACE2 transgene is driven by a non-native K18 promoter, allowing the infection of multiple organs, and resulting in tissue expression levels that are distinct from endogenously expressed murine Ace2[49,50]. Although the expression is non-physiological, we deliberately chose K18-hACE2 mice for this initial study of SARS-CoV-2 transmission in mice since our goal was to investigate and compare transmission of early and current pandemic isolates, some of which are unable to engage murine Ace2. We recognize that extrapolating tropism changes in our model to tropism in humans warrants caution, especially if these changes are solely brought upon by changes in variant spike-ACE2 binding, i.e., through mutations in spike's receptor binding site. However, we argue that tropism changes brought upon by differences in spike processing and/or by differences in host-virus interplay can be readily interpreted using our model, since protease expression and inflammation or antagonism are unadulterated by the hACE2 transgene. Taken together, our study demonstrates how the selection among available animal models should be based on research scope and context, as each of them can provide valuable and non-redundant information regarding SARS-CoV-2 biology.

In our model, except for Omicron BA.1, we observed at or above 33% SARS-CoV-2 variant transmission, and our method was granular enough to detect subtle differences between variants. For example, we identified an early but narrow peak of URT replication for Alpha, leaving only a short window for transmission. Despite this, contact animals acquire Alpha at a similar rate to ancestral WA-1, and even display significantly higher contact titers at 3 dpi. In fact, Alpha was the only virus with similar or better transmission efficiency to WA-1, while all other variants displayed a significant delay in transmission compared to WA-1. Variants post-Alpha arose after widespread population immunity brought by either prior infection or immunization, and viral evasion from neutralization due to mutations in the variants' spike gene is well-documented[64–70]. Thus, we speculate that transmission differences between WA-1 and variants Beta to Delta may have been less dramatic under adaptive immune pressure or even reveal a transmission advantage for these variants compared to the ancestral virus, to which most immunization regimens are currently matched. Future studies with this model will introduce adaptive immune pressures by vaccination of dams before pregnancy. Offspring will acquire

immunoglobulins via transplacental passage or milk, which has been shown to reduce transmission in our influenza A virus model[17].

SARS-CoV-2's accessory protein ORF8 is arguably the most enigmatic, as it only shares a 20% protein identity with SARS-CoV. For SARS-CoV-2, ORF8 is a secreted glycoprotein[40,71,72], and diverse immunomodulatory functions have been proposed, such as IL-17A mimicry[73,74], interference with the interferon type I pathway[43], histone mimicry[75], and downregulation of MHC class I[76,77], although the full spectrum of ORF8's functions and mechanisms of action remain to be elucidated[78]. In our model, deletion of ORF8 resulted in reduced replication specifically in the URT. The magnitude of inflammatory cytokines (Fig. 4) we observe for ΔORF8 in the URT was similar to those of parental WA-1 or ΔORF6 virus, despite 100-fold less virus in that compartment. This increased inflammation was similar to previous observations in adult K18-hACE mice[41,45]. It is feasible that, without ORF8, SARS-CoV-2 fails to suppress critical antiviral responses in the URT. We argue that the reduced URT replication and consequent decrease in shedding, is the cause of ΔORF8's delayed and reduced transmission in mice. We further argue that this is not due to an overall attenuation of SARS-CoV-2 lacking ORF8, as these viruses replicate efficiently in tissue culture and in the lungs of both neonatal and adult K18-hACE2 mice (Fig. 4 and ref. 45). To our knowledge, this is the to date the only report of a compartment-specific role of a SARS-CoV-2 accessory protein in mice. Although underlying mechanisms for the URT-specific role of ORF8 remain unknown, it is possible that antiviral inflammatory processes brought upon by SARS-CoV-2 infection are different in the URT from those in the lower respiratory tract. For instance, the cellular milieu and temperature differences between the URT and LRT may play a role in the behavior of the virus and response to infection in these compartments. Furthermore, the poor conservation of ORF8 among related coronaviruses[78], the appearance of a 382-nucleotide deletion in the ORF8 of SARS-CoV-2 isolates in patients during February 2020 and the circulation of a SARS-CoV-2 isolate containing a truncated version of ORF8 from March through October 2020[79] suggest that ORF8 is a hotspot for SARS-CoV-2 adaptation and evolution[78]. For example, the Alpha variant, which we also characterized in our study, carries a stop codon at amino acid 27 of ORF8, resulting in the expression of a truncated ORF8 protein[1]. Interestingly, in our model, ΔORF8 and Alpha did not phenocopy in terms of URT replication, cytokine responses, or transmission. It is possible that alpha's truncated ORF8 retains functions that are critical for URT replication. Another possibility is that alpha's ORF8 is defunct, but that other viral proteins, some of which are different from the WA-1 background, provide redundant immune suppression mechanisms that compensate for the lack of ORF8 action in Alpha. Future studies with additional recombinant SARS-CoV-2 will enable detangling the different hypotheses.

What are the determinants of SARS-CoV-2 transmission in our model? Our data suggest that the level of infectious index shedding titer matters with respect to transmission efficiency. We find that lower levels of index shedding are correlated with reduced transmission efficiency, as shown by both tested Omicron variants and by ΔORF8. Further, we observe that early timing of peak in index shedding (1 dpi vs 2 or 3 dpi) favors transmission efficiency, as shown by the slight transmission advantage of Alpha over WA-1 and the clear transmission advantage of Omicron BQ.1.1 over BA.1. Finally, we find that URT viral titers correlate with shedding titers across all viruses tested in this study, suggesting overall that efficient, early-onset URT replication determines SARS-CoV-2 transmission efficiency in our model. In contrast, in our model, we found that there was no association between the shedding of SARS-CoV-2 and the viral titers in the LRT, which suggests that the virus present in the lungs may be primarily responsible for causing disease, rather than facilitating its transmission.

In summary, we established a SARS-CoV-2 transmission model using neonatal K18-hACE2 transgenic mice that characterizes the net effect of variant-inherent mutations on viral tropism and transmission and uncovers the contribution of accessory proteins to transmission.

Using this tractable animal model to define the molecular mechanisms underlying SARS-CoV-2 transmission could guide the development of superior antiviral therapies and contribute to a greater understanding of not only SARS-CoV-2, but respiratory viruses in general.

## Methods

### Cell lines

Vero E6 cells were obtained from ATCC (CRL-1586) and cultured in Dulbecco modified Eagle medium (DMEM) (Gibco) supplemented with 10% fetal bovine serum (FBS) (Atlanta Biologicals), 1% Pen/Strep (Gibco) and 1% Amphotericin B (Gibco) at 37 °C with 5% CO2. Vero E6-TMPRSS2-T2A-ACE2 were obtained from BEI Resources (NR-54970, RRID: CVCL_C7NK) and cultured in DMEM (Corning) containing 4 mM L-glutamine, 4500 mg per L glucose, 1 mM sodium pyruvate and 1500 mg per L sodium bicarbonate, supplemented with 10% fetal bovine serum and 10 µg per mL puromycin (Sigma) at 37 °C with 5% $CO_2$. Both cell lines were confirmed to be mycoplasma free upon arrival and at monthly intervals.

### Mice

C57BL/6 J and K18-hACE2 C57BL/6 J (strain 2B6.Cg-Tg(K18-ACE2)2Prlmn/J) mice (Jackson Laboratories, ME) were maintained and bred in a conventional animal facility. To produce neonatal heterozygous K18-hACE2 C57BL/6 J mice for the transmission experiments, C57BL/6 J females were bred with homozygous K18-hACE2 C57BL/6 J males. Pups were housed with their mother during all experiments. Experimental animals of both sexes were used in all experiments and grouped together for analysis. Animal experiments were performed in the Animal Biosafety Level 3 (ABSL3) facility of NYU Grossman School of Medicine (New York, NY), in accordance with its Biosafety Manual and Standard Operating Procedures. The study received ethical approval by the NYU Grossman School of Medicine Animal Care and Use Committee (IACUC) under IACUC protocol # IA18-00071 (Dittmann).

### Biosafety and work with biohazardous material

All work with infectious SARS-CoV-2 isolates and recombinant viruses, including work with infected animals, was performed in the Animal Biosafety Level 3 (ABSL3) facilities of the University of Maryland (Baltimore, MD) and NYU Grossman School of Medicine (New York, NY). Both facilities are registered with their respective local Departments of Health and passed inspections by the Centers for Disease Control & Prevention (CDC (Centers for Disease Control)) within the year of submission of this manuscript. ABSL3 facilities are operated in accordance with their Biosafety Manuals and Standard Operating Procedures, including for the containment of biohazardous aerosols using certified biosafety cabinets and the facilities' sealed ventilation systems that provide a sustained directional airflow, from clean towards potentially contaminated areas, and HEPA-filtered exhaust. Biohazardous waste generated in the facilities is fully decontaminated using approved disinfectants followed by autoclaving and incineration as Regulated Medical waste. Access to ABSL3 facilities is restricted to certified and authorized personnel, enrolled into occupational health surveillance programs, and wearing adequate Personal Protective Equipment (PPE), including OSHA-approved respirators, eye protection, spill-resistant coveralls, and double-gloves. When analyzed outside of ABSL3 facilities, infectious samples were thoroughly treated using vetted inactivation methods.

All work with infectious SARS-CoV-2 isolates and recombinant viruses was performed with prior approval of the Institutional Biosafety Committees (IBCs) of the University of Maryland School of Medicine and NYU Grossman School of Medicine. Import permits for SARS-CoV-2 variant isolates were approved by the CDC. Generation of recombinant SARS-CoV-2 viruses was approved for M.F. by the IBC (Institutional Biosafety Committee) of the University of Maryland School of Medicine.

### Generation of Virus Stocks

The following reagents were obtained through BEI Resources, NIAID, NIH (National Institutes of Health): SARS-Related Coronavirus 2, Isolate USA-WA1/2020, NR-52281, deposited by the Centers for Disease Control and Prevention; SARS-Related Coronavirus 2, Isolate hCoV-19/England/204820464/2020, B.1.1.7, NR-54000, contributed by Bassam Hallis; SARS-Related Coronavirus 2, Isolate hCoV-19/South Africa/KRISP-EC-K005321/2020 (NR-54008), contributed by Alex Sigal and Tulio de Oliveira; SARS-Related Coronavirus 2, Isolate hCoV-19/Japan/TY7-503/2021 (Brazil P.1), NR-54982, contributed by National Institute of Infectious Diseases, SARS-CoV-2 Delta variant, isolate hCoV19/USA/PHC658/2021, B.1.617.2, NR-55611. SARS-CoV-2 Omicron BA.1 (hCoV-19/USA/GA-EHC-2811C/2021, EPI_ISL_7171744) were kindly provided by the Suthar Lab at Emory University. SARS-CoV-2 hCoV-19/USA/CA-Stanford-106_S04/2022 (Omicron BQ1.1, EPI_ISL_15196219) was obtained from Dr. Mehul Suthar (Emory University, Atlanta, Georgia, USA) and Dr. Benjamin Pinsky (Stanford University, Stanford, California, USA). The USA-WA1/2020 stock was produced as previously described[80]. The other SARS-CoV-2 viruses were passaged once in Vero E6 cells supplemented with 1 µg/ml of l-1-tosylamido-2-phenylethyl chloromethyl ketone (TPCK)-trypsin, to avoid virus adaptation to Vero E6 cells due to the lack of TMPRSS2 expression. Cells were infected at an MOI of 0.01 and harvested at 50% cytopathic effect (CPE). After harvest, the virus was purified using a 25% sucrose cushion at 25,000 RPM for 3–4 h and resuspended using PBS (Phosphate Buffered Saline) before infection. For the B.1.1.7, B.1.1.351 and P.1, stocks, aliquots were initially plaque-purified and sequenced to verify the variant signature before expanding in the presence of TPCK-trypsin to generate a passage 1 working stock. We perform a cellular debris exclusion step by benchtop centrifugation before proceeding with the virus pelleting step through a sucrose cushion. We then resuspend the pellet in a low volume, resulting in highly concentrated, purified SARS-CoV-2 stocks. WA-1 ΔORF6, WA-1ΔORF8 and its WA-1 control were generated in the Frieman Lab at University of Maryland School of Medicine[61]; aliquots were used as provided for the experiments. For heat-inactivated (HI) SARS-CoV-2 WA-1, viral stocks were incubated at 56 °C for 5 h. Samples were titered for confirmation of complete inactivation of infectious particles.

### Immunohistochemistry on mouse heads

Pups infected as described above were euthanized according to humane, IACUC-approved procedure at 2 dpi. The skin of heads was gently removed to preserve nasal structures. Heads were then removed and submerged in PBS at 4 °C for a brief wash, followed by fixing in 4% paraformaldehyde for 72 h at 4 °C without shaking. Heads were then washed in PBS at 4 °C with gentle swirling for 20 min, followed by decalcification into 0.12 M EDTA solution at 4 °C with gentle shaking for 7 days. Intact heads were then processed through graded ethanols to xylene and infiltrated with paraffin in a Leica Peloris automated tissue processor. Paraffin-embedded sections were immunostained on a Leica BondRX, according to manufacturer's instructions. In brief, deparaffinized sections underwent a 20-minute heat retrieval in Leica ER2 buffer (pH9, AR9640) followed by Rodent Block (Biocare, RBM961 L) before a 1-h incubation with SARS-CoV-2 N protein antibody (clone 1C7C7, Cell Signaling Technology, Cat #68344) at a 1:300 dilution and a AF594-conjugated Goat-anti-mouse secondary (ThermoFisher, Cat # A11005) at 1:100. Slides were counterstained with DAPI. Semi-automated image acquisition was performed on a Vectra® Polaris multispectral imaging system. After whole slide scanning at 20X the tissue was manually outlined to select fields for spectral unmixing and image analysis using InForm® version 2.6 software from Akoya Biosciences. Research image data was managed using OMERO Plus v5.6 (Glencoe Software) for viewing, annotation, and/or Figure making with OMERO.figure v 4.4 (OME team).

## SARS-CoV-2 infection of adult mice

Adult C57BL/6 J K18-hACE2$^{+/-}$ hemizygous mice (13 weeks of age, both sexes) were infected with a lethal dose (10,000 PFU) of ancestral SARS-CoV-2 USA_WA-1/2020 via a 10 μL intranasal infection under Ketamine/Xylazine anesthesia. Male and female mice were housed separately in groups of four at a 1:3 infected index to uninfected contact ratio. Shedding of virus was collected by dipping the nares of each mouse three times into viral medium (PBS plus 0.3% bovine serum albumin [BSA]) daily for 10 days. Intra-cage transmission was determined by infectious particle presence in contact mice. % Acquisition was scored and presented as described below. Moribund were euthanized by $CO_2$ asphyxiation followed by cervical dislocation when humane endpoint criteria were met. At 10 dpi, surviving animals were euthanized by $CO_2$ asphyxiation followed by cardiac puncture. Retro tracheal lavages of the upper respiratory tracts of surviving animals were conducted by pushing 500 μL of sterile PBS through the trachea and out the nares. Lungs were collected and homogenized with stainless steel beads as described below.

To assess seroconversion due to transmission, adult mice (13-week old) were infected as described above and housed separately in groups of 1:3 and 1:2 infected index: uninfected contacts. 13-week-old male mice were inoculated with 10 μL sterile PBS and 12-week-old female mice were inoculated with a 10 μL sublethal dose of SARS-CoV-2 USA_WA-1/2020 (1000 PFU). Mice were monitored daily for weight and survival. At 3 weeks postinfection, animals were euthanized by $CO_2$ asphyxiation followed by cardiac puncture. Retro tracheal lavage of the URT and lung homogenates were collected as described above. Blood was collected via cardiac puncture using a 1 mL 28G1/2 insulin syringe and added to a serum collection tube (BD Microtainer 365967). Tubes were incubated for 45 minutes at room temperature to facilitate blood coagulation and then spun at 4 °C for 10 minutes (1500 × g). Serum was separated, frozen at − 80 °C, and analyzed for presence of IgG antibodies against Spike Trimer (Acro Biosystems RAS-T023).

## SARS-CoV-2 infection of neonatal mice, determination of shedding and transmission

Neonatal mice were considered mice of 4–7 days of age. Sex of individual pups was not determined, and thus, animals of both sexes were likely used in all experiments with neonatal mice. Pups were infected with a 3 μL sterile PBS inoculum without general anesthesia (to avoid direct lung inoculation) by intranasal instillation of 1500 PFU of SARS-CoV-2 WA-1, SARS-CoV-2 variant or recombinant SARS-CoV-2 and returned to the nursing dam for the duration of the experiment. In transmission experiments, one or two pups, as indicated in the figure legend, were infected and returned to the (uninfected) littermates for the duration of the experiment. Shedding of virus was collected by dipping the nares of each mouse three times into viral medium (PBS plus 0.3% bovine serum albumin [BSA]) daily, and samples were evaluated via quantitative reverse transcription-PCR (RT-qPCR) or plaque assay. Intra-litter transmission was assessed by collecting shedding samples daily in the littermates (contact). % Acquisition was visualized in inverted Kaplan-Meier plots. Acquisition events were scored as at least two days of infectious viral titer in shedding samples. The onset of acquisition was scored as the first day of infectious virus detection. The pups and mother were euthanized by $CO_2$ asphyxiation followed by cardiac puncture. The upper respiratory tract (URT) was subjected to a retro tracheal lavage (flushing of 300 μL PBS from the trachea and collecting through the nares), and samples were used to quantify viruses (via plaque assay or qRT-PCR). Ratios of index to contact pups ranged from 1:6-9 for variant comparison to 1:3-1:4 for model optimization with WA-1.

## SARS-CoV-2 quantification RT-qPCR

RNA extractions were performed using the Qiagen QIAamp Viral RNA kit. The number of viral N copy number per μL was quantified by RT-qPCR using the Taqman® RNA-to-CT One-Step RT-PCR kit (Applied Biosystems™) with SARS-CoV-2 primers and probe targeting an amplicon in the N protein of SARS-CoV-2 WA-1 (Forward: 5'ATGCTGCAATCGTGCTACAA3'; reverse: 5' GACTGCCGCCTCTGCTC3') and the probe 5'/56-FAM/TCAAGGAAC/ZEN/AACATTGCCAA/3IABkFQ/3'. A standard curve was generated for each dataset using in vitro transcribed SARS-CoV-2 N RNA sequence (MN985325.1).

## SARS-CoV-2 infectious titer quantification

Infectious viral titers were determined by plaque assay. In brief, 10-fold dilutions of each virus in DMEM + 1% antibiotic/antimycotic (Gibco) were added to a monolayer Vero E6-TMPRSS2-T2A-ACE2 cells for 1 h at 37 °C. Following incubation, cells were overlaid with 0.8% agarose in DMEM containing 2% FBS and incubated at 37 °C for 36 hrs. Cells were fixed with 10% formalin, the agarose plug removed, and plaques visualized by crystal violet staining. Stocks and samples obtained from mouse experiments (shedding, lavage, and lung homogenates) were titered in Vero E6-TMPRSS2-T2A-ACE2.

## Lung SARS-CoV-2 titer quantification

Lungs were collected in 500 μL of DPBS containing one stainless steel bead (QIAGEN), homogenized with the Tissue-Lyser II (QIAGEN), and debris were pulled down at 8000 rpm for 8 min. Viral titers were determined by plaque assay using Vero E6-TMPRSS2-T2A-ACE2 cells.

## Cytokine and chemokine protein measurements

Cytokine and chemokine levels were measured in mouse serum using the Bio-Plex Pro Mouse Cytokine 23-plex Assay (Bio-Rad). Cytokines and chemokines were recorded on a MAGPIX machine (Luminex) and quantitated via comparison to a standard curve. xPONENT version 4.3.229.0 software was used for the data collection and analysis. We used the experimentally obtained limits of detection for each protein and set values below that limit of detection to one log below the limit. All samples were normalized to those from PBS-treated animals. Clustered heatmaps were generated using the ComplexHeatmap package v.2.14.0.

## Statistics and Reproducibility

All statistical analyses were performed using GraphPad Prism 9.5. Each experiment was completed at least two independent times with internal biological duplicates unless otherwise noted in the figure legends. Data are represented as the geometric mean ± geometric standard deviation (SD). Kruskal-Wallis and log-rank Mantel-Cox tests were performed and specifically indicated in the figure legends. Only p-values < 0.05 were displayed in the figures and p-values > 0.05 were considered non-significant (ns).

The exact number of biological replicates (n) is the following:
- Figure 1
  **b** index n = 4, contact n = 9
  **c** index n = 4, contact n = 9
  **d** index n = 4, contact n = 9
  **e** index n = 4, contact n = 9
  **f** n = 1 mouse
- Figure 2
  **b** n = 8 for longitudinal shedding, n = 7 for day 2 sampling
  **c** n = 15 for longitudinal shedding, n = 13 for day 2 sampling
  **d** n = 7 for longitudinal shedding, n = 5 for day 2 sampling
  **e** n = 8 for longitudinal shedding, n = 6 for day 2 sampling
  **f** n = 7 for longitudinal shedding, n = 5 for day 2 sampling
  **g** n = 12 for longitudinal shedding, n = 5 for day 2 sampling
  **h** n = 1 mouse per SARS-CoV-2 variant
- Figure 3
  **a** n = 12 for longitudinal shedding, n = 12 for Kaplan-Meier plots
  **b** n = 27 for longitudinal shedding, n = 27 for Kaplan-Meier plots
  **c** n = 14 for longitudinal shedding, n = 14 for Kaplan-Meier plots
  **d** n = 15 for longitudinal shedding, n = 15 for Kaplan-Meier plots

**e** n = 13 for longitudinal shedding, n = 13 for Kaplan-Meier plots
**f** n = 8 for longitudinal shedding, n = 8 for Kaplan-Meier plots
**g** n = 12 for longitudinal shedding, n = 12 for Kaplan-Meier plots
- Figure 4
**a** n = 7 for longitudinal shedding, n = 5 day 2 sampling
**b** n = 8 for longitudinal shedding, n = 6 day 2 sampling
**c** n = 10 for longitudinal shedding, n = 8 day 2 sampling
**d** n = 12 for longitudinal shedding, n = 12 for Kaplan-Meier plots
**e** n = 14 for longitudinal shedding, n = 14 day 2 sampling
**f** n = 13 for longitudinal shedding, n = 13 day 2 sampling
**g** n = 19 for individual URT samples (PBS n = 6, r-WA-1 n = 4, ORF6 n = 5, ORF8 n = 4), n = 21 for individual Lungs samples (PBS n = 6, r-WA-1 n = 5, ORF6 n = 5, ORF8 n = 5)
**h** n = 1 mouse per SARS-CoV-2 variant
- Supplemental Fig. 1
**b** weight: index n = 3 (m = 2, f = 1), contact n = 9 (m = 6, f = 3)
**c** survival curve: index n = 3 (m = 2, f = 1), contact n = 9 (m = 6, f = 3)
**d** longitudinal shedding: index n = 3 (m = 2, f = 1), contact n = 9 (m = 6, f = 3)
**e** endpoint lung titers: contact n = 9 (m = 6, f = 3)
**f** seroconversion (control and contact): n = 11 (PBS n = 3 (m = 3), sublethal n = 3 (f = 3), contacts n = 5 (m = 2, f = 3))
**h** weight: n = 12 (1500 PFU n = 6, 15000 PFU n = 3, 50000 PFU n = 3)
**i** mortality events: n = 21 (1500 PFU n = 9, 15000 PFU n = 9, 50000 PFU n = 3)
**j** longitudinal titers: n = 12 (1500 PFU n = 3, 15000 PFU n = 3, 50000 PFU n = 3)
**k** longitudinal shedding: n = 11
- Supplemental Fig. 2:
**a** longitudinal compartment titers: day 1: n = 5, day 2: n = 7, day 3: n = 4
**b** longitudinal compartment titers: day 1: n = 11, day 2: n = 13
**c** compartment titers: day 1: n = 6
**d** longitudinal shedding titers: n = 11
**e** longitudinal compartment titers: day 1: n = 5
**f** n = 1 mouse
- Supplemental Fig. 3:
**a** n = 12 for contacts, n = 2 for index
**b** n = 27 for longitudinal shedding, n = 4 for index
**c** n = 14 for longitudinal shedding, n = 2 for index
**d** n = 15 for longitudinal shedding, n = 2 for index
**e** n = 13 for longitudinal shedding, n = 2 for index
**f** n = 8 for longitudinal shedding, n = 2 for index
**g** n = 12 for longitudinal shedding, n = 2 for index
- Supplemental Fig. 4:
**a** n = 19 for individual URT samples (PBS n = 6, r-WA-1 n = 4, ORF6 n = 5, ORF8 n = 4), **b** n = 21 for individual Lungs samples (PBS n = 6, r-WA-1 n = 5, ORF6 n = 5, ORF8 n = 5)
- Supplemental Figure 5:

**a** r-WA-1 longitudinal weight: index n = 4, contact n = 16, longitudinal survival (deaths): index: n = 3, contact n = 14
**b** ORF6 longitudinal weight: index n = 2, contact n = 14, longitudinal survival (deaths): index: n = 2, contact n = 7
**c** ORF8 longitudinal weight: index n = 2, contact n = 13, longitudinal survival (deaths): index: n = 2, contact n = 6

## Reporting summary

Further information on research design is available in the Nature Portfolio Reporting Summary linked to this article.

## Data availability

All data generated in this study are provided in the Supplementary Information/Source Data file. Source data are provided with this paper.

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

## Acknowledgements

We thank Thomas M. Moran, Icahn School of Medicine at Mount Sinai, and Luis Martínez-Sobrido, Texas Biomedical Institute, for the kind gift of mouse monoclonal SARS-CoV N antibody 1C7, and Mehul Suthar, Emory School of Medicine, and Benjamin Pinsky, Stanford University, for the gift of the variants. We thank Ashley Fisher and Nicole Rondeau for their assistance in assembling the Source Data file. We are grateful to the SARS-CoV-2 Assessment of Viral Evolution (SAVE) Program for critical input on our data. Histopathology of the murine nasopharynx was performed by Mark Alu and Branka Brukner Dabovic from the NYUMC Experimental Pathology Research Laboratory (RRID: SCR_017928). Both the Experimental Pathology Research Laboratory and the NYU Genome Technology Core are supported by NYU Cancer Center support grant P30CA016087 and by NYU Langone's Laura and Isaac Perlmutter Cancer Center and the Vectra Polaris multispectral scanner was purchased through a Shared Instrument Grant S10 OD021747. We are also grateful to the NYU Langone Antimicrobial-Resistant Pathogens (AMR) Program and thank Adriana Heguy and the team from the NYU Genome Technology Core for deep sequencing of all viruses used in this study. Research was partially supported by the following grants from NIH/NIAID: R01AI143639 to M.D., K08AI141759 to M.B.O, R01AI143861 to K.M.K., DK093668 to K.C., and T32AI007180. None of the work with recombinant SARS-CoV-2 at NYU Grossman School of Medicine was funded by NIAID. Work was further supported by The Vilcek Institute of Graduate Biomedical Sciences, and by NYU Grossman School of Medicine Startup funds.

## Author contributions

B.A.R.-R.: Conceptualization, Methodology, Validation, Formal analysis, Investigation, Writing – Original draft, Visualization. G.O.C.: Conceptualization, Methodology, Validation, Formal analysis, Investigation, Writing – Review & Editing, Visualization. R.D.: Conceptualization, Methodology, Validation, Formal analysis, Investigation, Writing – Review & Editing, Visualization. A.M.V.-J.: Conceptualization, Methodology, Investigation, Writing – Review & Editing. S.T.Y.: Methodology, Investigation, Writing – Review & Editing. K.M.C.: Conceptualization, Methodology, Investigation, Writing – Review & Editing. A. R.S.: Methodology, Investigation, Writing – Review & Editing. L.B.-R.: Conceptualization, Methodology, Investigation, Writing – Review & Editing. J.J.R.G.: Methodology, Investigation, Writing – Review & Editing. M.E. M.: Methodology, Resources, Writing – Review & Editing. S.V.: Methodology, Resources. Yong Xue: Methodology, Resources. Cynthia Loomis: Methodology, Supervision. K.M.K.: Methodology, Supervision, Funding Acquisition. Kenneth Cadwell: Methodology, Supervision, Funding Acquisition. Ludovic Desvignes: Methodology, Resources, Supervision, Writing – Review & Editing. Matthew B. Frieman: Conceptualization, Methodology, Resources, Supervision, Funding Acquisition, Writing – Review & Editing. Mila B Ortigoza: Conceptualization, Methodology, Formal analysis, Investigation, Writing – Original draft, Visualization, Supervision, Funding Acquisition. Meike Dittmann: Conceptualization, Methodology, Formal analysis, Investigation, Writing – Original draft, Visualization, Supervision, Funding Acquisition.

## Competing interests

The authors declare no competing interests.

## Additional information

[1]Department of Microbiology, New York University Grossman School of Medicine, New York, NY 10016, USA. [2]Department of Medicine/Division of Infectious Diseases and Immunology, New York University Grossman School of Medicine, New York, NY 10016, USA. [3]Vaccine Center, NYU Grossmann of Medicine, New York, NY 10016, USA. [4]Department of Microbiology and Immunology, Center for Pathogen Research, University of Maryland School of Medicine, Baltimore, MD 21201, USA. [5]Department of Synthetic Biology and Bioenergy, J. Craig Venter Institute, Rockville, MD 20850, USA. [6]Department of Pathology, New York University Grossman School of Medicine, New York, NY 10016, USA. [7]Perlmutter Cancer Center, New York University Langone Health, New York, NY 10016, USA. [8]Division of Gastroenterology and Hepatology, Department of Medicine, University of Pennsylvania Perelman School of Medicine, Philadelphia, PA 19104, USA. [9]Department of Systems Pharmacology and Translational Therapeutics, University of Pennsylvania Perelman School of Medicine, Philadelphia, PA 19104, USA. [10]Department of Pathology and Laboratory Medicine, University of Pennsylvania Perelman School of Medicine, Philadelphia, PA 19104, USA. [11]High Containment Laboratories - Office of Science and Research, NYU Langone Health, New York, NY 10016, USA. [12]These authors contributed equally: Bruno A. Rodriguez-Rodriguez, Grace O. Ciabattoni. [13]These authors jointly supervised this work: Mila B Ortigoza, Meike Dittmann.
✉e-mail: Mila.Ortigoza@nyulangone.org; Meike.Dittmann@nyulangone.org

