## [Peer Review File · Nature Communications]

A neonatal mouse model characterizes transmissibility of SARS-CoV-2 variants and reveals a role for ORF8Reviewers' Comments:

Reviewer #1:

Remarks to the Author:

The novel coronavirus SARS-CoV-2, which emerged in 2019, spread worldwide and caused a pandemic while giving rise to variant strains with increased transmission in human societies. Animal models of transmission of SARS-CoV-2 are important in elucidating the scientific background of the increased (in human society) transmissibility of the many variants that emerged. It would also be very significant if it reflected the actual transmission of the virus in human society.

In this paper, Bruno et al. present a small animal (mouse) model to analyze the transmissibility of SARS-CoV-2. The animals used were neonatal mice genetically modified to express the human receptor ACE2. Using this model, the authors systematically analyzed the early epidemic strain and several variants that subsequently became predominant and replaced the previous strain (and may have increased its ability to spread in human society).

The authors conducted their experiments very carefully and meticulously with these animals, and their data would be reliable. However, the analysis searching for molecular bases underlying the different transmissibility among variants was limited in content.

If this model reflects to some extent the transmissibility of SARS-CoV-2 (and its variants) in human society, it must be said that the transmissibility of the omicron variant is remarkably low. In this regard, the omicron variant has always been an exception to the experimental results, and given that the virus strain currently prevalent in human society is the omicron strain, the question remains to what extent this animal model reflects the transmissibility of SARS-CoV-2 (and its variants) to humans.

1. Why are quantitative values shown for many of the analytical data, such as viral titers, even for results below the LOD?
2. In addition to the analysis of variants, the analysis of recombinant viruses deficient in ORF6 and ORF8 seems abrupt. At the very least, the basic properties of these recombinant viruses should be shown using cultured cells.
3. Authors say that they confirmed the nucleotide sequence of the virus stocks used in the experiments by deep sequencing, but what did they confirm by deep sequencing to determine that the virus stocks used in the experiment were appropriate (line 145)?
4. What did authors compare the sequence data to to determine that there was no mutation due to culture (line 146)?
5. Did authors confirm the absence of cell debris (degree of purification of virus particles) by electron microscopy or other means (line 147)?
6. Authors speculated that the cause of the early decline in viral titer in mice for specific variants was a decline in stability, but what kind of change in viral particles do authors consider (lines 153 and 183)?
7. Authors used VeroE6 cells and VerE6-TMPRSS2-T2A-ACE2 cells in different experiments. Each variant has been shown to have different infectivity to VeroE6 cells and TMPRSS2-expressing cells; does the use of these cells affect the results or conclusions of this study? It may be necessary to re-measure the infectivity titer of the stocks used in the analysis with TMPRSS2-expressing Vero cells as well.
8. RNA levels are also decreasing (line 126).
9. A decreasing trend in titer is also observed (line 128).

Reviewer #2:

Remarks to the Author:

Rodriguez-Rodriguez et al. report a neonatal mouse model for characterizing the transmissibility of SARS-CoV-2 variants and ORF6 and ORF8 deletion viruses. The study provides an interesting and useful small animal model for SARS-CoV-2 research. The overall study was well performed. The following suggestions could substantiate the manuscript.

Major points

1. Since this is a study of model development, the authors should provide kinetics of viral loads in URT and lungs on different days post infection in index pups.
2. The authors should perform experiments to examine the viral stability to support their speculation indicated in lines 182-184.

Minor points

1. Legend to Fig. 3f is missing.
2. Line 54 add reference doi: 10.1016/j.celrep.2020.108234
3. Line 65, add reference doi: 10.1038/s41467-022-31930-z

Reviewer #3:

Remarks to the Author:

This manuscript demonstrates the transmission of multiple non-species-adapted SARS-CoV-2 variants in neonatal C57BL/6 K18-hACE2 mice, which do not support virus transmission when mature. The experiments seem straightforward with sound methods, and the authors' conclusions are generally appropriate, although they could perhaps be worded more circumspectly. Specifically, I would caution the authors not to over-interpret the relevance of their findings to humans quite yet. This animal model is still in its infancy (pun intended). Mice are an altricial species, and neonatal mice are less mature than even newborn babies, not to mention older humans. For instance, stating that this manuscript is the "first report of a compartment-specific role of a SARS-CoV-2 accessory protein" (line 384) is technically true but could also be qualified with the words "in mice," "in this model," or the like added at the end of the sentence. Same goes for "Our study ... provides evidence of an accessory protein, ORF8, to be critical for SARS-CoV-2 transmission" (line 90). ORF8 may not have a compartment- or transmission-specific role in any other species, or even in older mice for that matter, so statements of this sort would be optimally transparent if they were qualified accordingly.

Additionally, I'm not sure that one can automatically conflate peak nasal shedding titers with transmissibility, at least not under all circumstances. For one thing, I'm not sure that the possibility of transmission by short-range aerosol particles has been ruled out in this model. (See comment below, line 68). If that is the case, LRT exhaled viral load may be more important than nasopharyngeal titers for aerosol transmission (for influenza virus, anyway; see Yan et al., Proc Natl Acad Sci USA 2018;115(5):1081-1086 PMID: 29348203), and exhaled viral load may not correlate with the URT titer (Port et al., bioRxiv 2022.08.15.504010; doi: 10.1101/2022.08.15.504010). Again extrapolating from the flu literature, virus transmission also seems to be at least as dependent on when peak shedding occurs as on how high it is (Danzy et al., J Virol. 2021 Mar 17;95(11): e02320-20, PMID: 33731462; Mubareka et al., J Infect Dis 2009;199(6):858-65, PMID: 19434931). While not specifically designed to test this observation, "transmission chain" experiments perhaps show qualitatively similar transmission kinetics occurring with SARS-CoV-2; as the peak titer in donor hamsters "shifts to the right," transmission probability decreases (Fig 5A, Port et al.; doi: 10.1101/2022.08.15.504010), just as it does with influenza (Danzy et al.). I think that probably more work needs to be done in this model to understand, first of all, what transmission mechanism(s) are in play, and, second, whether degree of nasal shedding does indeed always neatly correlate with transmissibility.

SPECIFIC COMMENTS

Line 68, "Ferrets display ... minimal aerosol transmission": Just a word of caution here, that transmission frequency is not mistaken for transmission mechanism. By necessity, to demonstrate

transmission via aerosol particles, the index/infected animal must be a good distance away from the naïve/uninfected sentinel animal, or the airflow between them must be made to go through impingers, around corners, or other impediments that weed out the larger particles. The experimental setup serves to dilute the concentration of viable airborne virus that even reaches the partner animal, such that the index animal must be emitting a lot of viruses in order for a critical mass of them to make it through this experimental obstacle course and deliver an infectious dose to the naïve animal. It is entirely possible that, even in a contact-permissive setting like the infant mouse model, that transmission between pups is occurring primarily via aerosol (meaning, via very small airborne particles that can be inhaled deep within the respiratory tract) – albeit via very short-range aerosol transmission. Because the index and sentinel are so close together, the volume of air between them is very small, and thus the concentration of virus that reaches the sentinel remains high; very little dilutional loss of virus can occur in such a small volume of air. Just because the pups are wriggling all over each other and mom doesn't mean that the virus is spreading via a mechanism involving direct/indirect contact or short-range droplet sprays. It's just that one cannot tease out very short-range aerosol transmission from the other mechanisms in a contact-permissive experimental setup. Obviously, the frequency of sentinel infection via an aerosol mechanism is going to be higher when the volume of air between sentinel and index is very small (i.e., when they are close together) than when it is very large (i.e., the complicated aerosol-only setup used by Kutter et al. [Nat Commun. 2021 Mar 12;12(1):1653, PMID: 33712573]), simply because the virus becomes less concentrated as it travels towards the sentinel in the latter scenario. But importantly, these differences in transmission frequency tell us nothing about the mechanism by which the virus is transmitting. It could be telling us that the index ferret is just not always emitting enough virus to ensure that an infectious dose's-worth is making it through the obstacle course to the sentinel ferret in the other cage. It doesn't mean that aerosol transmission (again, meaning transmission via very small, inhalable airborne virus particles) would be "minimal" if index and sentinel ferrets were able to be closer to each other.

Line 71, "...aerosol transmission in most experimental hamster scenarios is 100% effective, which complicates the assessment [of/for?] increased transmission": Missing word.

Line 111, "We monitored the pup's weight and survival...": Pups' (possessive plural).

Lines 121 and 146, "inoculates": I believe that "inoculate" is only a verb. The noun form with which I am most familiar is "inoculum/a".

Line 217, "NIAID SAVE investigators": Acronyms may not be known to all readers.

Line 221, "Using omicron as a non-transmitter, we define that index shedding titers at or below 2.7×10^3 PFU/mL would likely not allow for transmission in our model": To be precise, one can only really define the shedding titer transmission threshold with a virus that does transmit in this model, for instance by altering the conditions of the index infection to decrease peak titers to a level below which the normally transmitting virus no longer transmits. It is entirely possible that the transmission bottleneck with omicron is not just the peak titer it can achieve. Some other viral attribute may also (or instead) be preventing efficient transmission in this model. (See for instance Mubareka et al., PMID: 19434931, for a flu virus that achieves a high peak titer but does not transmit efficiently in guinea pigs.)

Line 242, "For equally replicating viruses WA-1, alpha, beta, gamma, and delta we found that cytokine signatures did not generally correlate with their transmission dynamics": I'm no immunologist, but I found this observation interesting, in light of the hypothesis that neonatal mice transmit viruses that older mice do not due in part to their relative immune immaturity or tolerance towards infections. The fact that variants with different transmission dynamics induced similar cytokine expression profiles suggests either that induction of innate immunity does not play a significant role in transmission dynamics (or at least induction of the cytokines assessed) or that other variant-specific virus-host interactions also modulate the transmission-permissiveness of neonatal mice.

Line 269, "Next, we observed a trend of higher recombinant SARS-CoV-2 WA-1 titers in the lungs, 5×10^5 PFU/mL, and lower titers in the URT and shedding samples, at 1×10^5 or 5×10^4 PFU/mL, respectively (Fig 4d)": It's not clear what the rWA1 titers are being compared to – wtWA-1? If the comparison is rWA-1 titers in the LRT vs. URT/shedding, then it would be clearer to use "than" instead of "and" ("...higher recombinant SARS-CoV-2 WA-1 titers in the lungs, 5×10^5 PFU/mL, THAN in the URT and shedding samples..."). "And" makes it read as if LRT and URT titers are both being compared to something else (i.e., "...rWA-1 achieved higher titers in the lung and lower titers in the URT, relative to an unspecified comparator...").

Line 327, "We argue that this route of infection [unanesthetized intranasal inoculation] may recapitulate SARS-CoV-2's natural infection route better than models using deep anesthesia, in which deep inhalation favors lower respiratory tract infection. Although inoculation with anesthesia has been shown to produce the typical lung pathology associated with COVID-19 in K18-hACE2 mice, this approach could have a significant impact on transmission as it mainly circumvents the initial URT infection": I'm not aware of any hard evidence that SARS-CoV-2 initiates infection in the nose/URT in humans, although I have often seen it stated (unreferenced, of course). There is circumstantial evidence – for instance, receptor expression levels are highest in the nose and decrease deeper into the respiratory tract, and ex vivo ciliated cells seem to be a preferred target cell type for SARS-CoV-2 – but as far as I know, no one has ever shown direct evidence that infection initiates above the larynx and then extends below the larynx via inhalation or aspiration of progeny virus – as logical as that hypothesis is. Physiologically speaking, could not infection initiate in, say, the ciliated cells of the intrapulmonary airways and then be spread upwards, above the larynx, by exhalation or coughing? Milton's group has shown for influenza virus (granted, a different virus) that humans experimentally infected via intranasal inoculation while awake are different, symptomatically and physiologically, from those who were naturally infected in the community (Bueno de Mesquita et al., *Influenza Other Respi Viruse*. 2021; 15: 154-163), suggesting that intranasal inoculation in general is not mimicking a natural infection route in humans (again, at least for flu).

Line 338, "For instance, our transmission model is based on a mixture of droplet and contact transmission": I do not think that ref. 40 or the present manuscript conclusively rule out the possible involvement of a short-range aerosol transmission mechanism in this model; see above comment on line 68.

Line 368, "immunization regiments": Regimens, I think?

REVIEWER COMMENTS

Reviewer #1 (Remarks to the Author):

The novel coronavirus SARS-CoV-2, which emerged in 2019, spread worldwide and caused a pandemic while giving rise to variant strains with increased transmission in human societies. Animal models of transmission of SARS-CoV-2 are important in elucidating the scientific background of the increased (in human society) transmissibility of the many variants that emerged. It would also be very significant if it reflected the actual transmission of the virus in human society. In this paper, Bruno et al. present a small animal (mouse) model to analyze the transmissibility of SARS-CoV-2. The animals used were neonatal mice genetically modified to express the human receptor ACE2. Using this model, the authors systematically analyzed the early epidemic strain and several variants that subsequently became predominant and replaced the previous strain (and may have increased its ability to spread in human society). The authors conducted their experiments very carefully and meticulously with these animals, and their data would be reliable. However, the analysis searching for molecular bases underlying the different transmissibility among variants was limited in content. If this model reflects to some extent the transmissibility of SARS-CoV-2 (and its variants) in human society, it must be said that the transmissibility of the omicron variant is remarkably low. In this regard, the omicron variant has always been an exception to the experimental results, and given that the virus strain currently prevalent in human society is the omicron strain, the question remains to what extent this animal model reflects the transmissibility of SARS-CoV-2 (and its variants) to humans.

1. Why are quantitative values shown for many of the analytical data, such as viral titers, even for results below the LOD?

Thank you for bringing this up. We have now defined in each figure legend how we included data points below the LOD. For viral titers, the LOD of our plaque assay is 50 PFU/ml, and we now set all negative data to one log below that, which is 5. For ELISAs, we used the defined experimentally for individual runs and for each individual protein and set negative data one log below that individual LOD. Of note, mean values may still appear below the LOD if individual values pull the mean below the LOD.

2. In addition to the analysis of variants, the analysis of recombinant viruses deficient in ORF6 and ORF8 seems abrupt. At the very least, the basic properties of these recombinant viruses should be shown using cultured cells.

Thank you for raising this point. The recombinant viruses have been previously characterized in-depth by our co-authors in cultured cells regarding their replication kinetics, and in adult mice regarding their pathogenesis (McGrath et al., PNAS 2022). They had not been characterized regarding their transmission potential. While we referenced that previous publication in the first submission, we apologize for not making this point clearer. We now emphasize the previous characterization and discuss their data in view of the data provided in our study in more depth.

3. Authors say that they confirmed the nucleotide sequence of the virus stocks used in the experiments by deep sequencing, but what did they confirm by deep sequencing to determine that the virus stocks used in the experiment were appropriate (line 145)? What did authors compare the sequence data to determine that there was no mutation due to culture (line 146)?

Thank you for asking this question. We indeed have a rigorous stock generation and stringent quality control pipeline in place to produce clonal virus with confirmed sequence identity (i.e. presence of variant-defining mutations) and integrity (i.e. absence of artifactual mutations introduced by virus propagation in cell culture). The comparator sequences used were openly available sequences of the initial isolate for each referenced variant (see Methods section). All stocks used in this study have been produced this way (Steps 1-8, Figure below). A manuscript on our method is currently in revision at *Nature Protocols*. We will cite this protocol in the final version of the neonatal manuscript.

4. Did authors confirm the absence of cell debris (degree of purification of virus particles) by electron microscopy or other means (line 147)?

While we did not perform EM to confirm the absence of cell debris, sucrose ultracentrifugation has been shown to purify a variety of viruses, enveloped and non-enveloped, including other viruses of the coronavirus family (Arora et al., 1985; Mbiguino and Menezes, 1991; Putnak et al., 1996; Ali and Roossinck, 2007; Hankaniemi et al., 2017; Leibowitz et al., 2011). We perform a cellular debris exclusion step by benchtop centrifugation prior to proceeding with the virus pelleting step through a sucrose cushion. We then resuspend the pellet in a low volume, resulting in highly concentrated, purified SARS-CoV-2 stocks. We added this to the Methods section.

5. Authors speculated that the cause of the early decline in viral titer in mice for specific variants was a decline in stability, but what kind of change in viral particles do consider (lines 153 and 183)?

We removed this statement from the manuscript.

6. Authors used VeroE6 cells and VeroE6-TMPRSS2-T2A-ACE2 cells in different experiments. Each variant has been shown to have different infectivity to VeroE6 cells and TMPRSS2-expressing cells; does the use of these cells affect the results or conclusions of this study? It may be necessary to re-measure the infectivity titer of the stocks used in the analysis with TMPRSS2-expressing Vero cells as well.

Thank you for raising this comment. We now reiterated the titers in both on Vero E6 and on VeroE6-TMPRSS2-T2A-ACE2 cells. We found that, across variants, the “conversion rate” from Vero E6 to VeroE6-TMPRSS2-T2A-ACE2 titers in our hands was about 50-fold. This is consistent with findings by others. Further, VeroE6-TMPRSS2-T2A-ACE2-determined inocula presented very similar across variants, at 1500 PFU/3 μ l inoculum. We thus changed the information on calibrated inocula from 30 PFU (previously on Vero E6) to 1500 PFU (now on VeroE6-TMPRSS2-T2A-ACE2). We also changed the Methods section to reflect that these VeroE6-TMPRSS2-T2A-ACE2 cells were used for all plaque assays in this study.

7. RNA levels are also decreasing (line 126).

Thank you, we modified this sentence.

8. 9A decreasing trend in titer is also observed (line 128).

Thank you, we modified this sentence.

Reviewer #2 (Remarks to the Author):

Rodriguez-Rodriguez et al. report a neonatal mouse model for characterizing the transmissibility of SARS-CoV-2 variants and ORF6 and ORF8 deletion viruses. The study provides an interesting and useful small animal model for SARS-CoV-2 research. The overall study was well performed. The following suggestions could substantiate the manuscript.

1. Since this is a study of model development, the authors should provide kinetics of viral loads in URT and lungs on different days post infection in index pups.

Thank you for this fantastic suggestion. We performed a time course of viral loads from shedding samples, URT lavages, and lungs for the WA-1 ancestral strain. The data is now included in Extended Data Figure 2a. Additionally, we included day 1 and day 2 compartment data for Alpha and Omicron BQ.1.1, which peak earlier than WA-1 (Extended Data Figure 2b, c).

2. The authors should perform experiments to examine the viral stability to support their speculation indicated in lines 182-184.

We decided to delete this speculation from the manuscript.

3. Legend to Fig. 3f is missing.

Thank you, 3f was added to the legend.

4. Line 54 add reference doi: 10.1016/j.celrep.2020.108234

Thank you, the reference was added.

5. 3. Line 65, add reference doi: 10.1038/s41467-022-31930-z

Thank you, the reference was added.

Reviewer #3 (Remarks to the Author):

This manuscript demonstrates the transmission of multiple non-species-adapted SARS-CoV-2 variants in neonatal C57BL/6 K18-hACE2 mice, which do not support virus transmission when mature. The experiments seem straightforward with sound methods, and the authors' conclusions are appropriate, although they could be worded more circumspectly.

1. Specifically, I would caution the authors not to over-interpret the relevance of their findings to humans quite yet. This animal model is still in its infancy (pun intended). Mice are an altricial species, and neonatal mice are less mature than even newborn babies, not to mention older humans. For instance, stating that this manuscript is the “first report of a compartment-specific role of a SARS-CoV-2 accessory protein” (line 384) is technically true but could also be qualified with the words “in mice,” “in this model,” or the like added at the end of the sentence. Same goes for “Our study ... provides evidence of an accessory protein, ORF8, to be critical for SARS-CoV-2 transmission” (line 90). ORF8 may not have a compartment- or transmission-specific role in any other species, or even in older mice for that matter, so statements of this sort would be optimally transparent if they were qualified accordingly.

We thank the reviewer for this comment and acknowledge our overstatements. We changed language as suggested throughout the manuscript.

2. Additionally, I'm not sure that one can automatically conflate peak nasal shedding titers with transmissibility, at least not under all circumstances.

We agree with this statement and now phrase this point more carefully. We also added two correlation graphs (Extended Data Fig. 2g and Extended Data Fig. 4d) and added a paragraph to the discussion about determinants of transmission in our model.

3. For one thing, I'm not sure that the possibility of transmission by short-range aerosol particles has been ruled out in this model. (See comment below, line 68).

We agree, and acknowledged in the discussion that modes of transmission cannot be tested experimentally in this model.

4. If that is the case, LRT exhaled viral load may be more important than nasopharyngeal titers for aerosol transmission (for influenza virus, anyway; see Yan et al., Proc Natl Acad Sci USA 2018;115(5):1081-1086 PMID: 29348203), and exhaled viral load may not correlate with the URT titer (Port et al., bioRxiv 2022.08.15.504010; doi: 10.1101/2022.08.15.504010).

We agree with this fully. We and others have shown previously that influenza virus URT titers do not correlate with transmission, but shedding does. We have added the suggested references and added this statement to the discussion.

5. Again extrapolating from the flu literature, virus transmission also seems to be at least as dependent on when peak shedding occurs as on how high it is (Danzy et al., J Virol. 2021 Mar 17;95(11): e02320-20, PMID: 33731462; Mubareka et al., J Infect Dis 2009;199(6):858-65, PMID: 19434931).

We agree fully, and have made similar observations in the neonate model with influenza virus. We now highlight this point more strongly, and added the suggested citations.

6. While not specifically designed to test this observation, “transmission chain” experiments perhaps show qualitatively similar transmission kinetics occurring with SARS-CoV-2; as the peak titer in donor hamsters “shifts to the right,” transmission probability decreases (Fig 5A, Port et al.; doi: 10.1101/2022.08.15.504010), just as it does with influenza (Danzy et al.). I think that probably more work needs to be done in this model to understand, first of all, what transmission mechanisms are in play, and, second, whether degree of nasal shedding does indeed always neatly correlate with transmissibility.

Unfortunately, due to the fact of working with suckling mice, detangling the exact modes of transmission in our model is not technically possible, which we now also expanded upon in the discussion. We also added a paragraph to the discussion about determinants of transmission in our model.

7. Line 68, “Ferrets display ... minimal aerosol transmission”: Just a word of caution here, that transmission frequency is not mistaken for transmission mechanism.

We agree and reworded that sentence, as well as the following sentence regarding hamsters.

8. By necessity, to demonstrate transmission via aerosol particles, the index/infected animal must be a good distance away from the naïve/uninfected sentinel animal, or the airflow between them must be made to go through impingers, around corners, or other impediments that weed out the larger particles. The experimental setup serves to dilute the concentration of viable airborne virus that even reaches the partner animal, such that the index animal must be emitting a lot of viruses in order for a critical mass of them to make it through this experimental obstacle course and deliver an infectious dose to the naïve animal. It is entirely possible that, even in a contact-permissive setting like the infant mouse model, that transmission between pups is occurring primarily via aerosol (meaning, via very small airborne particles that can be inhaled deep within the respiratory tract) – albeit via very short-range aerosol transmission. Because the index and sentinel are so close together, the volume of air between them is very small, and thus the concentration of virus that reaches the sentinel remains high; very little dilutional loss of virus can occur in such a small volume of air. Just because the pups are wriggling all over each other and mom doesn’t mean that the virus is spreading via a mechanism involving direct/indirect contact or short-range droplet sprays.

We fully agree with these statements.

9. It’s just that one cannot tease out very short-range aerosol transmission from the other mechanisms in a contact-permissive experimental setup. Obviously, the frequency of sentinel infection via an aerosol mechanism is going to be higher when the volume of air between sentinel and index is very small (i.e., when they are close together) than when it is very large (i.e., the complicated aerosol-only setup used by Kutter et al. [Nat Commun. 2021 Mar 12;12(1):1653, PMID: 33712573]), simply because the virus

becomes less concentrated as it travels towards the sentinel in the latter scenario. But importantly, these differences in transmission frequency tell us nothing about the mechanism by which the virus is transmitting. It could be telling us that the index ferret is just not always emitting enough virus to ensure that an infectious dose's-worth is making it through the obstacle course to the sentinel ferret in the other cage. It doesn't mean that aerosol transmission (again, meaning transmission via very small, inhalable airborne virus particles) would be "minimal" if index and sentinel ferrets were able to be closer to each other.

We fully agree with these statements.

10. Line 71, "...aerosol transmission in most experimental hamster scenarios is 100% effective, which complicates the assessment [of/for?] increased transmission": Missing word.

Thank you, this has been fixed.

11. Line 111, "We monitored the pup's weight and survival...": Pups' (possessive plural).

Thank you, this sentence has been deleted in the revised version.

12. Lines 121 and 146, "inoculates": I believe that "inoculate" is only a verb. The noun form with which I am most familiar is "inoculum/a".

Thank you, this has been fixed.

13. Line 217, "NIAID SAVE investigators": Acronyms may not be known to all readers.

Thank you, the data on the impairment of omicron BA.1 to transmit via aerosols in hamsters is now published and referenced as such.

14. Line 221, "Using omicron as a non-transmitter, we define that index shedding titers at or below 2.7×10^3 PFU/mL would likely not allow for transmission in our model": To be precise, one can only really define the shedding titer transmission threshold with a virus that does transmit in this model, for instance by altering the conditions of the index infection to decrease peak titers to a level below which the normally transmitting virus no longer transmits. It is entirely possible that the transmission bottleneck with omicron is not just the peak titer it can achieve. Some other viral attribute may also (or instead) be preventing efficient transmission in this model. (See for instance Mubareka et al., PMID: 19434931, for a flu virus that achieves a high peak titer but does not transmit efficiently in guinea pigs.)

This point is well-taken, and we added it to the discussion. We performed this experiment, but found that titrating the virus inoculum below 1500 PFU yields inefficient infection of index, such that some pups have productive infection whereas others are not infected at all. Thus, we did not include the data in the revised manuscript.

15. Line 242, "For equally replicating viruses WA-1, alpha, beta, gamma, and delta we found that cytokine signatures did not generally correlate with their transmission dynamics": I'm no immunologist, but I found this observation interesting, in light of the hypothesis that neonatal mice transmit viruses that older mice do not due in part to their relative immune immaturity or tolerance towards infections. The fact that variants

with different transmission dynamics induced similar cytokine expression profiles suggests either that induction of innate immunity does not play a significant role in transmission dynamics (or at least induction of the cytokines assessed) or that other variant-specific virus-host interactions also modulate the transmission-permissiveness of neonatal mice. Difficult to know without further experiments in future projects.

Thank you so much for raising this point. Based on your comment, we revisited our cytokine data and now also included new data on the dynamics of cytokine levels post infection over time. Doing so, we made the following observations: 1. robust cytokine induction for most variants occurs at the 48 h time point and requires active viral replication; 2. infection with ancestral virus and variants (except omicron, which does not replicate well in mice) triggers a similar set of cytokines, but the extent of cytokine induction varies, 3. high cytokine induction is roughly associated with better transmission for most variants, except Alpha (Extended Data Fig 4d). Future experiments with transgenic mice carrying interruptions in key inflammatory pathways, i.e. *lfnar1*^{-/-}, will enable us to test this potential association in mechanistic detail.

16. Line 269, “Next, we observed a trend of higher recombinant SARS-CoV-2 WA-1 titers in the lungs, 5x10⁵ PFU/mL, and lower titers in the URT and shedding samples, at 1x10⁵ or 5x10⁴ PFU/mL, respectively (Fig 4d)”: It’s not clear what the rWA1 titers are being compared to –wtWA-1? If the comparison is rWA-1 titers in the LRT vs. URT/shedding, then it would be clearer to use “than” instead of “and” (“...higher recombinant SARS-CoV-2 WA-1 titers in the lungs, 5x10⁵ PFU/mL, THAN in the URT and shedding samples...”). “And” makes it read as if LRT and URT titers are both being compared to something else (i.e., “...rWA-1 achieved higher titers in the lung and lower titers in the URT, relative to an unspecified comparator...”).

Thank you, this sentence has been modified accordingly, and nomenclature of “WA-1” (isolate) and “rWA-1” (recombinant) has been changed throughout the manuscript.

17. Line 327, “We argue that this route of infection [unanesthetized intranasal inoculation] may recapitulate SARS-CoV-2’s natural infection route better than models using deep anesthesia, in which deep inhalation favors lower respiratory tract infection. Although inoculation with anesthesia has been shown to produce the typical lung pathology associated with COVID-19 in K18-hACE2 mice, this approach could have a significant impact on transmission as it mainly circumvents the initial URT infection”: I’m not aware of any hard evidence that SARS-CoV-2 initiates infection in the nose/URT in humans, although I have often seen it stated (unreferenced, of course). There is circumstantial evidence – for instance, receptor expression levels are highest in the nose and decrease deeper into the respiratory tract, and ex vivo ciliated cells seem to be a preferred target cell type for SARS-CoV-2 – but as far as I know, no one has ever shown direct evidence that infection initiates above the larynx and then extends below the larynx via inhalation or aspiration of progeny virus – as logical as that hypothesis is. Physiologically speaking, could not infection initiate in, say, the ciliated cells of the intrapulmonary airways and then be spread upwards, above the larynx, by exhalation or coughing?

Milton's group has shown for influenza virus (granted, a different virus) that humans experimentally infected via intranasal inoculation while awake are different, symptomatically and physiologically, from those who were naturally infected in the community (Bueno de Mesquita et al., *Influenza Other Respi Viruse*. 2021; 15: 154-163), suggesting that intranasal inoculation in general is not mimicking a natural infection route in humans (again, at least for flu).

Thank you for pointing this out. We have now taken out this sentence, as we indeed cannot make comparisons to human progression of infection. However, we now included new data showing that, in our model, WA-1 infection progresses from the upper to the lower respiratory tract over time (Extended Data Fig 2a).

18. Line 338, "For instance, our transmission model is based on a mixture of droplet and contact transmission": I do not think that ref. 40 or the present manuscript conclusively rule out the possible involvement of a short-range aerosol transmission mechanism in this model; see above comment on line 68.

We agree and removed this sentence.

19. Line 368, "immunization regiments": Regimens, I think?

Thank you, this has been corrected.

Reviewers' Comments:

Reviewer #1:

Remarks to the Author:

The authors need to answer the following comments (questions) of the first round review.

' If this model reflects to some extent the transmissibility of SARS-CoV-2 (and its variants) in human society, it must be said that the transmissibility of the omicron variant is remarkably low. In this regard, the omicron variant has always been an exception to the experimental results, and given that the virus strain currently prevalent in human society is the omicron strain, the question remains to what extent this animal model reflects the transmissibility of SARS-CoV-2 (and its variants) to humans.'

Regarding the comment 1:

It is not critical for assessing the importance of this study, I doubt that the method of evaluating data below the detection limit with a 1-log lower value is generally acceptable.

Regarding the comment 6:

The authors re-measured the infectious titer of viral stocks in TMPRSS2-expressing cells and found that the difference was equally 50-fold for all viral strains (an early epidemic strain and all variants) and changed the description in the paper accordingly. However, early epidemic strains are known to be less dependent on TMPRSS2, while VOCs (except for the Omicron variant) have an increased capacity to utilize TMPRSS2. As a result, it has been reported that in early epidemic strains, titer measurements in VeroE6 cells approximate those in TMPRSS2-expressing VeroE6 cells, whereas in many VOCs, titers in TMPRSS2-expressing VeroE6 cells are greatly increased. In other words, the viral titer ratio in VeroE6 cells and TMPRSS2-expressing Vero cells can be different for each virus strain, especially between the early endemic strain and VOCs. The method of using VeroE6 cells to determine the titer of the stock virus and the amount to be inoculated into mice according to that titer can be considered feasible, but the result that all virus strains are equally 50-fold is questionable.

Reviewer #3:

Remarks to the Author:

I thank the authors for their attention to my comments and concerns, which have been satisfactorily addressed.

RESPONSE TO REVIEWERS' COMMENTS

Reviewer #1 (Remarks to the Author):

The authors need to answer the following comments (questions) of the first round review.

'If this model reflects to some extent the transmissibility of SARS-CoV-2 (and its variants) in human society, it must be said that the transmissibility of the omicron variant is remarkably low. In this regard, the omicron variant has always been an exception to the experimental results, and given that the virus strain currently prevalent in human society is the omicron strain, the question remains to what extent this animal model reflects the transmissibility of SARS-CoV-2 (and its variants) to humans.'

Thank you for bringing this up – we had actually made some changes in the manuscript to address this but failed to add it to the rebuttal letter. We agree that initially, the low transmissibility of Omicron in our system struck us as unusual. However, since we made our initial findings, the body of work demonstrating the incompatibility of Omicron with rodents has been growing. This also pertains to reduced transmission between rodents, as demonstrated in the hamster model. We place our findings in perspective of published literature (lines 328-333). We also discuss how our model's limitations and how it may relate to SARS-CoV-2 transmission in human in depth in the discussion (lines 478-486).

Regarding the comment 1:

It is not critical for assessing the importance of this study, I doubt that the method of evaluating data below the detection limit with a 1-log lower value is generally acceptable.

How to depict undetectable values / values below the LOD in scientific papers remains a topic of debate. =LOD/2, =LOD/squareroot(2), =LOD/10, =LOD, =0, =1, or recommendations of additional statistical analysis of which value to use for a specific dataset can be found in the literature, including Nature family journals. There are no editorial guidelines on this. We define both the LOD and the depiction of values <LOD, and use this consistently for each panel of the manuscript.

Regarding the comment 6:

The authors re-measured the infectious titer of viral stocks in TMPRSS2-expressing cells and found that the difference was equally 50-fold for all viral strains (an early epidemic strain and all variants) and changed the description in the paper accordingly. However, early epidemic strains are known to be less dependent on TMPRSS2, while VOCs (except for the Omicron variant) have an increased capacity to utilize TMPRSS2. As a result, it has been reported that in early epidemic strains, titer measurements in VeroE6 cells approximate those in TMPRSS2-expressing VeroE6 cells, whereas in many VOCs, titers in TMPRSS2-expressing VeroE6 cells are greatly increased. In other words, the viral titer ratio in VeroE6 cells and TMPRSS2-expressing Vero cells can be different for each virus strain, especially between the early endemic strain and VOCs. The method of using VeroE6 cells to determine the titer of the stock virus and the amount to be inoculated into mice according to that titer can be considered feasible, but the result that all virus strains are equally 50-fold is questionable.

In response to the initial question, we had re-titrated inoculates on VeroE6-ACE2-TMPRSS2 and VeroE6 cells in parallel. While WA-1 appeared higher on VeroE6-ACE2-TMPRSS2 than other viruses, all other variants were equally at 1500 PFU/3 μ l (volume of neonate inoculum). Given that the VeroE6 titers varied more among each other, and that we measure all other virus titers in this manuscript on VeroE6-ACE2-TMPRSS2, we feel that indicating VeroE6-ACE2-TMPRSS2 titer as 1500 PFU inoculum is the right choice.

Reviewer #3 (Remarks to the Author):

I thank the authors for their attention to my comments and concerns, which have been satisfactorily addressed. We thank reviewer 3 for their valuable input, which we feel has made the manuscript stronger.